



# Sesquiterpenes and oxygenated sesquiterpenes dominate the emissions of downy birches

Heidi Hellén[1], Arnaud P. Praplan[1], Toni Tykkä[1], Aku Helin[1], Simon Schallhart[1], Piia P. Schiestl-Aalto[2,3,4], Jaana Bäck[2,3], Hannele Hakola[1]

[1]Finnish Meteorological Institute, P.O. Box 503, 00101 Helsinki, Finland
[2]Institute for Atmospheric and Earth System Research / Forest Sciences
[3]Faculty of Agriculture and Forestry, University of Helsinki, Finland
[4]Department of Forest Ecology and Management, SLU, 901 83 Umeå, Sweden

*Correspondence to:* Heidi Hellén (heidi.hellen@fmi.fi)

**Abstract.** Even though isoprene and monoterpene (MT) emissions of boreal needle trees have been studied quite intensively, there is less knowledge on the emissions of broadleaved deciduous trees and emissions of larger terpenes and oxygenated volatile organic compounds (OVOCs). Here we studied the downy birch (*Betula pubescens*) leaf emissions of terpenes, OVOCs and green leaf volatiles (GLVs) at the SMEAR II boreal forest site using in situ gas chromatographs with mass spectrometers in 2017 and 2019.

The highest emissions were detected during the early growing season, indicating that bud break and early leaf growth are a strong source of these compounds. Sesquiterpenes (SQTs) and oxygenated sesquiterpenes (OSQTs) were the main emitted compounds almost throughout the summer. Mean emissions (averaged over bud break/early/main and late growing season) of SQTs and OSQTs were $5 - 690$ and $46 - 650$ ng $g_{dw}^{-1}$ $h^{-1}$, respectively. Isoprene emissions were very low or below detection limits (seasonal means $< 0.4$ ng $g_{dw}^{-1}$ $h^{-1}$), whereas variable levels of emissions of MTs, $C_5$-$C_{10}$ aldehydes and GLVs were detected. On average SQT and OSQT emissions were 5 and 6 times higher than MT emissions, but variation over the growing season was high and during the late growing season MTs were the main compound group emitted.

Of the SQTs, β-caryophyllene and β-farnesene were the main compounds emitted in 2019, while in 2017 also high, possibly stress-induced emissions, of α-farnesene were detected. The main emitted OSQTs were tentatively identified as 14-hydroxy-β-caryophyllene acetate (M 262 g/mol) and 6-hydroxy-β-caryophyllene (M 220 g/mol). Of the MTs, α-pinene, β-pinene, limonene and sabinene were the most abundant compounds except during the last two days of the measurements in August in 2019, when β-ocimene emissions had the major contribution. In 2017, when the measured tree was suffering from leaf damages possibly due to drought and high chamber temperature, high emissions of GLVs, linalool, α-farnesene and an unidentified SQT were detected.

Emission potentials calculated for 30ºC had very high variation between seasons and the highest potentials were detected during bud break and early growing season. Emission potentials calculated separately for each measurement day followed the leaf growth and were highest during the most rapid leaf growth.

To our knowledge this is the first time when birch emission rates of OSQTs have been quantified. Even with low emissions, these compounds are expected to have strong impacts on the atmospheric chemistry and especially on secondary organic aerosol (SOA) production.





## 1 Introduction

Monoterpenes (MTs) and sesquiterpenes (SQTs) are the major biogenic volatile organic compounds (BVOCs) emitted from the boreal forest. Both groups exhibit many structural isomers, with a large range of reactivity. They influence

chemical communication of plants and insects, the oxidation capacity of air and particle formation and growth. BVOC emissions are known to be highly dependent on temperature and light (Guenther et al. 2012), but also other abiotic stress factors such as frost, drought, radiation and exposure to oxidants such as ozone (Vickers et al. 2009, Loreto and Schnizler 2010, Bourtsoukidis et al. 2012 and 2014b) may have strong impacts on the emissions. In addition, biotic stress factors such as herbivore/pathogen outbreaks can initiate or alter VOC emissions (Pinto-Zevallos et al. 2013,

Joutsensaari et al. 2015, Faiola and Taipale 2020).

Several studies have shown that there is a lot of unknown hydroxyl radical reactivity in the air especially in boreal forests (Yang et al., 2016). For example, in a forest in Finland the unknown fraction of reactive compounds is over 50% (Nölscher et al., 2012, Praplan et al., 2019). Missing reactivity has been suggested to originate from unknown

emitted compounds or atmospheric oxidation products of VOCs. A recent study (Praplan et al. 2019) showed that currently known oxidation products are able to explain only a minor fraction (< 4.5%) of the missing reactivity in the air of boreal forest and large fractions of missing reactivity, which was not explained by isoprene, MTs or SQTs, have been found directly in the emissions of main boreal tree species (Nölcher et al. 2013, Praplan et al., 2020). To explain this unknown reactivity, we need to search for new compounds.

Although isoprene and MTs have been studied quite intensively in boreal areas (e.g. Hakola et al., 2001, 2006, 2017; Hellén et al. 2006, 2018, Mäki et al. 2019; Kesselmeier et al., 1999; Rinne et al., 2007, 2009; Ruuskanen et al., 2007; Tarvainen et al., 2005) and there are also studies on sesquiterpenes (Bourtsoukidis et al., 2014; Hakola et al., 2001, 2006, 2017, Mäki et al. 2017, Hellén et al., 2018), knowledge on the emissions of BVOCs other than terpenes is very

limited. Emissions of small oxygenated VOCs (OVOCs) like methanol, acetone and acetaldehyde are well characterized (e.g. Mäki et al. 2019, Aalto et al. 2014, Bourtsoukidis et al. 2014a, Schallhart et al. 2018), but there is very little information on the emissions of higher OVOCs. König et al. (1995) found significant oxygenated hydrocarbon emissions from Silver birch (*Betula pendula*), Hakola et al. (2017) found emissions of $C_5$-$C_{10}$ aldehydes from Norway spruces and Wildt et al. (2003) detected stress induced emissions of $C_5$-$C_{10}$ aldehydes from 6 different

plant species in their laboratory experiments. To our knowledge there are no publications on birch emission rates of oxygenated sesquiterpenes (OSQTs). Zang et al. (1999) detected them in the head space of Downy birch branches and Isidorov et al. (2019) in the head space of collected birch buds. They are also known to be a major component in the essential oils of birches (Dimirce et al. 2000, Klike et al. 2004). Earlier they have been detected in the emissions of desert and Mediterranean shrubs and some trees growing in the warmer vegetation zones (Matsunaga et al. 2009,

Yaman et al. 2015 and Yanez-Serrano et al. 2018). These OSQTs are expected to be highly reactive and to have higher comparative secondary organic aerosols yields than isoprene and MTs and therefore even low emissions may have strong impacts in the atmosphere.



There are many studies on BVOC emissions of the main coniferous trees in boreal forests, but the data on deciduous trees is more limited (Guenther et al., 2012). Birches are common deciduous broadleaved trees in northern and central Europe and in Asia.In Finland, downy birch (*Betula pubescens*) is the most common broadleaved tree. Downy and silver birch emissions in natural environment have been studied by Hakola et al. (1998, 2001), and König et al. (1995) and mountain birch (subspecies of Downy birch) emissions by Ahlberg (2011) and Haanpanala et al. (2009). They have found highly variable emissions of both MTs and SQTs.

Here we set up to study the downy birch emissions of terpenes, OVOCs and green leaf volatiles (GLVs) at the SMEAR II boreal forest site using in situ gas chromatographs with mass spectrometers (GC-MSs). To our knowledge this is the first time when emission rates of OSQTs have been quantified in Downy birch emissions.

## 2    Methods

### 2.1 Studied trees

Downy birch (*Betula pubescens* Ehrh.) emission measurements were conducted at the SMEAR II station (Station for Measuring Forest Ecosystem–Atmosphere Relations; 61º51'N, 24º18'E; 181 a.s.l.) in Hyytiälä, southern Finland (Hari and Kulmala, 2005). In 2017, a young downy birch (height of about 2 m) was planted in a 10-liter pot with a mixture of mineral soil and fertilized growth peat and placed next to the measurement container. The seedling received normal rainwater and in drier periods it was watered. In 2019, a downy birch (height ~4 m) growing naturally next to the container was measured.

Measurements were conducted at different stages of the leaf development (Table 1). In 2017 during the early growing season measurement period (24–28/5/2017) leaves of the downy birch were growing fast (Fig. 1). During the main growing season (samples taken 21–28/6/2017 and 13–19/7/2017) leaves were fully grown and some leaf damages were detected. Visible signs of senescence of the leaves started during the late growing season (samples taken 23-28/8/2017).

In 2019, growth of the leaves started a bit earlier than in 2017. In 2019, samples were taken already during the bud break period between 7 and 15 May ('bud'). Leaves were growing fast during the early growing season (samples taken during the period 18/5–7/6/2019, 'early'). During the main growing season (samples taken during the periods 18/6–5/7/2019 and 21–27/7/2019, 'main') leaves were fully grown and in the end of the late growing season (samples taken during the period 10–23/8/2019, 'late') senescence of leaves was starting. The two first measurements ('bud' and 'early') were conducted from the same branch, and in the following periods two new branches were measured, so that altogether three branches were used in the measurements in both years

Table 1. Sampling schedule showing the measurement periods during each stage (bud/early/main/late) of the growing season.



| Year | Bud | Early | Main | Late |
|------|-----|-------|------|------|
| 2017 | n.a. | 24/–28/5 | 21–28/6 | 23–28/8 |
| | | | 13–19/7 | |
| 2019 | 7–15/5 | 18/5–7/6 | 18/6–5/7 | 10–23/8 |
| | | | 21–27/7 | |

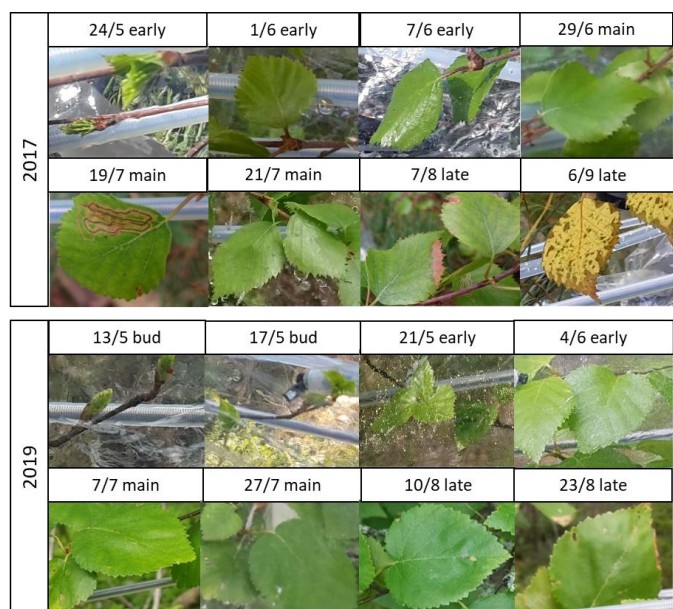

Figure 1: The measured leaves over the growing seasons in 2017 and 2019

## 2.2 Branch chamber measurements

### 2.2.1 Measurement set up

For the measurements, the downy birch branches were placed in a FEP enclosure (~6L cylinder) and the emission rates were measured using a steady state flow-through technique (Hakola et al. 2017). The FEP cover was removed after each measurement period and pulled back again at least 30 min before the next measurement, allowing the branch to experience ambient conditions in between. A zero air generator (HPZA-7000, Parker Hannifin Corporation) was used to produce the air flushed into the chamber (the flow rate was recorded continuously, and it was approximately 2.9 – 5.3 L min$^{-1}$ and 6.0 – 7.1 L min$^{-1}$ in 2017 and 2019, respectively). Due to the high reactivity and short atmospheric lifetimes of the studied compounds, their ambient air concentrations are much lower than concentrations in our




emission chamber. Therefore, we do not expect any artificial emissions by using zero air. However, in 2017, due to dry zero air, relative humidity of the air in the branch chamber was very low (mean 32±9 %). In addition, during that year the birch was growing in a pot and even though it was watered regularly, it is possible that there were occasional short drought episodes. In 2019, the in-going zero air was humidified with ultrapure water, and the mean relative

humidity of the air in the chamber was 66±19 %. The relative humidity (RH) and the temperature in the enclosure were recorded with a thermistor (Philips KTY 80/110, Royal Philips Electronics, Amsterdam, Netherlands) and the photosynthetically active radiation (PAR) was measured with a quantum sensor (LI-190SZ, LI-COR, Biosciences, Lincoln, USA) placed on top of the enclosure. The main flow going to the instruments was approximately at flow rate of 0.8 L min$^{-1}$ and 1.5 – 4.7 L/min in 2017 and in 2019, respectively. The main sampling line was a FEP tubing (ca. 5

m length, i.d. 1/8″) in 2017. In 2019, the main sampling line was a heated FEP tubing (ca. 10 m length, i.d. 1/8″) in all other periods except on 25 – 27/7, during which the sampling line was an unheated FEP tubing (ca. 13.5 m length, 3/8" i.d). The target compound losses in these sampling lines and enclosure are expected to be negligible as demonstrated by the acceptable recoveries observed in the laboratory tests (Helin et al., 2020; Hellén et al., 2012), and since high flow rates were used.

In order to determine the emission rates per leaf biomass, the measured branches were cut, dried at +60℃ and the leaves were weighted.

### 2.2.2 Temperature effect

The temperature inside the branch chamber is known to increase more than ambient temperatures in sunny conditions (e.g. Ortega and Helmig, 2008; Rinnan et al., 2014). Higher temperature is expected to induce more emissions. The flow through the chamber during the measurements was maintained at minimum 3 L min$^{-1}$ to keep the leaf surface temperature close to the chamber air temperature. However, the temperature of the leaves is also known to be higher than ambient air temperature in sunny conditions. To characterize this we measured surface temperature of the leaves

during the emission measurements using an infrared thermometer (IRT 206) in July/August 2019. Measured leaves were from the branches next to our branch chamber.

### 2.3 GC-MS analysis

Emissions of MTs, SQTs, oxygenated MTs (OMTs), oxygenated SQTs (OSQTs), isoprene, C$_4$-C$_{10}$ aldehydes (ALD) and green leaf volatiles (GLVs) were measured from branch chambers with two in situ thermal desorption-gas chromatograph-mass spectrometers (TD-GC-MSs).

In 2017 and in 2019, one instrument (GC-MS1) was used for the measurements of individual MTs, SQTs, isoprene

and OMTs (1,8-cineol, linalool, α-terpineol). Additionally in 2017, ALDs, and in 2019 OSQTs and *cis*-3-hexen-1-ol were measured with this instrument. Measured compounds are listed in Table A1 in the appendix A. VOCs were collected in the cold trap (Tenax TA 60-80/ Carbopack B 60-80 and Tenax TA 60-80 in 2017 and 2019, respectively) of the thermal desorption unit (TurboMatrix 350, Perkin-Elmer) connected to a gas chromatograph (Clarus 680, Perkin-Elmer) coupled to a mass spectrometer (Clarus SQ 8 T, Perkin-Elmer). To remove humidity, the cold trap was kept at





25°C during sampling. Between the 2017 and 2019 measurements, the thermal desorption unit was modified to enable the measurements of less volatile compounds, namely OSQTs. Stainless steel lines inside the online box of the TD unit were changed to FEP tubing and an empty sorbent tube used in the TD inlet line was changed to glass coated stainless steel tube. The optimization of the method is described in Helin et al. (2020). These TD unit changes and the use of a

GC column with a lower film thickness enabled the measurement of OSQTs. A HP-5 column (60 m, id. 0.25 mm, film thickness 1 µm, from Agilent Technologies) and a Elite-5MS column (60 m, id. 0.25 mm, film thickness 0.25 µm, from Perkin-Elmer) were used for separation in 2017 and 2019, respectively. To calibrate compounds other than isoprene, standards were injected as methanol solutions into sorbent tubes (Tenax TA 60-80/Carbpack B 60-80) and the tubes were thermally desorbed and analyzed as samples. Five-point calibration curves were used. To calibrate

isoprene, a gas standard from National Physical Laboratories (UK) was used instead of a methanol solution. Blank values measured by sampling an empty cuvette were subtracted from the results. SQTs (α-farnesene and SQT1-7), OSQTs (OSQT1-9) and MTs (β-ocimene and sabinene) lacking authentic standards were tentatively identified based on the comparison of the mass spectra and retention indexes (RIs) with NIST mass spectral library (NIST/EPA/NIH Mass Spectral Library, version 2.0). These tentatively identified compounds were quantified using response factors of

calibrated compounds having the closest RI and mass spectra resemblance. Regarding OSQTs, an authentic standard was available only for caryophyllene oxide. 30-minute samples at a flow rate of 40 mL min$^{-1}$ were collected every hour or every other hour. The total number of samples were 407 and 1102 in 2017 and 2019, respectively.

In 2017, an additional instrument (GC-MS2) was used for the measurements of GLVs. Measured compounds are listed

in Table A2 in the appendix A. Samples were analyzed in situ with a thermal desorption unit (Unity 2 + Air Server 2, Markes International LTD) connected to a gas chromatograph (Agilent 7890A, Agilent Technologies) and a mass spectrometer (Agilent 5975C, Agilent Technologies). A polyethylene glycol column, the 30 m DB-WAXetr (J&W 122-7332, Agilent Technologies, Santa Clara, CA, USA, i.d. 0.25mm, film thickness 0.25 µm) was used for the separation. Samples were taken every other hour. The sampling time was 60 min and the flow rate was 40 mL min$^{-1}$.

The method has been described in Hellén et al. (2017 and 2018).

### 2.4 Defining the growth of the leaves

Daily growth rate of the leaves, $G^i$, was modelled with the CASSIA model (Schiestl-Aalto et al., 2015) where the daily growth is driven by environmental variables, mainly daily mean temperature. CASSIA has previously been

parameterized for Scots pine at the SMEAR II station, and the model has been shown to successfully predict the growth of different tree organs (Schiestl-Aalto et al., 2015). Here, the model was parameterized for birch with leaf growth measured from photographs taken three times per week during summers 2015 and 2016. Furthermore, the parameter defining the timing of bud break was fitted to the observed bud break of our measurement branches. The mass of growing leaves on day $d$ is then $W_d = \sum_{i=1}^{d} G^i$ and the relative leaf mass on day $d$ is $W_d^r = W_d/W_e$ where $W_e$ is the

leaf mass on the last day of the measurement period.

### 2.5 Sampling of additional trees



To screen the birch individuals for their potential chemodiversity, additional samples from 1–5 m tall downy and silver birches growing close to the measurement container were taken in 2019. Branches were closed into a FEP-bags and 5-minute samples with a flow of approx. 200 ml min⁻¹, with the exact flow logged at the beginning and the end of sampling, were collected from the head space of the birch branches into the Tenax TA(60-80)-Carbopack B(60-80) sorbent tubes. The sample tubes were analysed later in the laboratory using a similar method as for the online samples (offline sorbent tube analysis, e.g. Helin et al. (2020)). These measurements were semi-quantitative and they were only used for qualitative analysis of terpene emission patterns. Since the concentrations in the FEP-bags were clearly higher than the ambient air concentrations, no subtraction of the initial concentration level was conducted.

### 2.6 Calculation of emission potentials

It is well-known that monoterpene emissions from coniferous boreal forests have an exponential temperature dependence (Guenther et al. 2012). The Guenther algorithm was used for calculating the emission potentials at 30°C (Guenther et al., 1993):

$$E = E_{30} \times \exp(\beta\,(T - T_S))$$

where $E_{30}$ is the standard emission potential at 30 °C (ng $g_{dw}^{-1}$ h⁻¹), T is the chamber temperature (°C), $T_S$ is the standard temperature (30 °C) and $\beta$ is the temperature sensitivity (°C⁻¹) of emissions. In addition to seasonal mean emission potentials, daily mean emission potentials ($E_{30}$) were calculated for 2019 for all days with more than half of the measurements available. To test if some of the emissions had delayed temperature effect emissions were also correlated with 2 hour time lag of temperature.

BVOC emissions may also be correlated with the level of irradiation, thus the emission potentials for light and temperature dependent emission ($E_{CTxCL}$) were also calculated with an algorithm developed by Guenther et al. (1993 and 1995). The temperature inside the chamber and the PAR measured on top of the chamber were used for these calculations. The standard emission potential can be obtained by linearly fitting the emission rates to the light and temperature activity factors (CTxCL) of the emission algorithm. Emission potentials ($E_{CT\,x\,CL}$) were defined at Ts 30°C and PAR 1000 µmol m⁻² s⁻¹. Unfortunately, the PAR sensor was not working/installed properly during the bud break season in 2019, thus the $E_{CTxCL}$ could not be calculated for this period.

### 3 Results and discussion

### 3.1 Emission rates of BVOCs

The highest total BVOC emissions from downy birch leaves were detected during the early growing season (Fig. 2 and Appendix A Table A1), i.e. from leaves that were still growing. High early growing season emissions indicate that these emissions are related to bud break/early growth of leaves, which has previously been shown, e.g., in Scots pine foliage (Aalto et al 2014). The SQTs and OSQTs were clearly the main compound groups emitted. On average SQT





emissions were 5 and 4 times higher than MT emissions in 2017 and 2019, respectively. OSQT emissions (measured only in 2019) were 5.8 times higher than MT emissions. Emissions of $C_4$-$C_{10}$ aldehydes (measured only in 2017) were as low or lower than MT emissions. In 2017, when GC-MS2 was used for measuring GLVs, high emission rates were observed simultaneously with leaf damages.

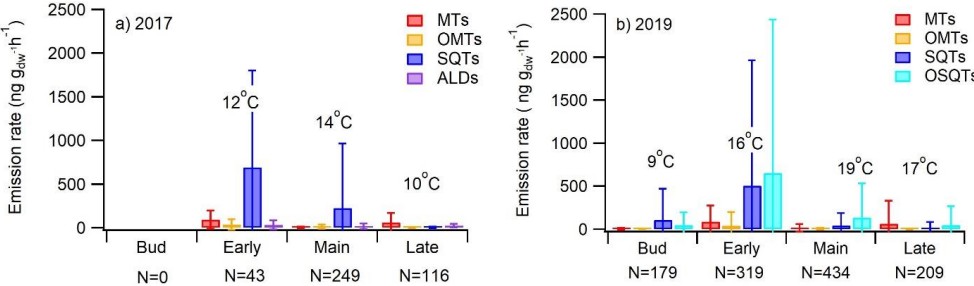

Figure 2: Season (bud break, early, main, late) mean± standard deviation emission rates of different VOC groups and mean chamber temperature in a) 2017 and b) 2019. N is the number of samples. ALD=$C_4$-$C_{10}$ aldehydes measured only in 2017.

Emissions were peaking in the afternoon, coinciding with the highest temperature and PAR, and were very low during the night (Fig 3). During the early growing season, when emissions were the highest, also some nighttime emissions were detected, indicating that the emissions are not just light dependent, but there are also emissions from storage pools
20   in downy birch. This was the case particularly in SQT and OSQT emissions. Due to clearly higher emissions in the afternoon and short lifetime of the terpenes, they are expected to have major effect on atmospheric chemistry especially during that time of the day.

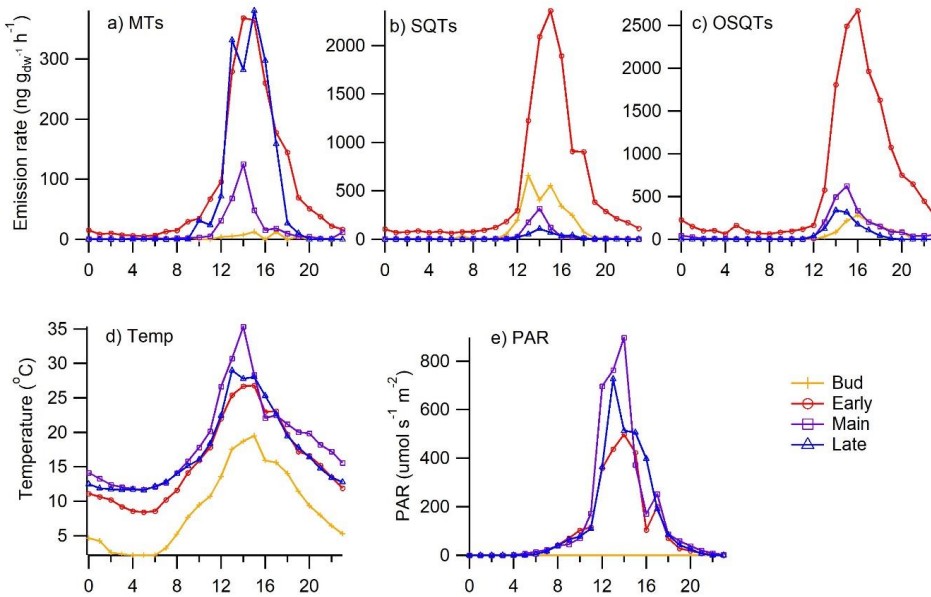

Figure 3: Diurnal variation of emission rates of a) MTs, b) SQTs and c) OSQTs and d) chamber temperature and e) PAR during different seasons in 2019. During bud break PAR was not measured.

### 3.1.1 Monoterpenoid emissions

MT emission rates were the highest during the early growing season, decreased during the main growing season and peaked again at the end (Fig. 2 and 3), which was comparable to a previous study from downy birch (e.g. Hakola et al

10    2001). Of the MTs, α-pinene, β-pinene, limonene and sabinene were the most abundant compounds except during the last two days of the measurements in August in 2019, when β-ocimene emissions had the major contribution (Fig. 4a). This could indicate that ocimene emissions are related to the senescence of the leaves. Our finding supports the previous observations, e.g. the ones from Vuorinen et al. (2005) where (E)-β-ocimene and (Z)-ocimene were main compounds emitted by the Silver birches in August. In addition to late growing season emissions, Vuorinen et al. (2007) found

15    strong herbivore induced emissions of ocimenes from Silver birches.

Linalool, one of the OMTs, had the highest emissions during the early growing season (Table A1 in the appendix A and Fig. 4b). In 2017, linalool had also relatively high emissions in June and July while it was not detected at all at the end of the growing season in August. In 2019, linalool was detected only during the early growing season, similarly

20    as reported by Hakola et al. (2001) who detected high linalool emissions from downy birch also in early growing season. However, linalool emission is also known to be induced by stress in Norway spruce (Petterson, 2007; Blande et al., 2009). The potential relationship of linalool with birch stress emissions is addressed later in Section 3.4.



### 3.1.2 SQT emissions

SQT emissions from the downy birch were clearly higher than MT emissions, except in the end of the growing season (Fig. 2). Seasonal mean SQT emission rates were 5 – 690 and 14 – 505 ng $g_{dw}^{-1}$ $h^{-1}$ in 2017 and 2019, respectively, being clearly the highest during the early growing season. High SQT emissions have been observed also in earlier studies of downy birch (Hakola et al., 2001, Räsänen et al., 2017) and mountain birch (*Betula pubescens* ssp. czerepanovii), which is a subspecies of downy birch (Haapanala et al., 2009).

In 2017, individual SQTs emission pattern changed dramatically over the growing season (Fig. 4c). In May, when the leaves were growing, β-caryophyllene had the highest contribution, while in the end of the growing season, it was not detected at all. α-Farnesene had very low emissions in May, while during the other seasons, it was the major SQT detected. In July 2017, emissions of an unidentified SQT (SQT7) were even higher than emissions of α-farnesene, while in May and June it was negligible. Since the birch in 2017 was likely suffering from leaf damage, drought and high chamber temperatures, it is possible that SQT7 and α-farnesene are stress related emissions (see Section 3.4). Hakola et al. (2001) found high emissions of β-caryophyllene from all downy birches, but α-farnesene was emitted only from young trees. In 2019, α-farnesene and SQT7 were not detected and β-caryophyllene, β-farnesene and α-humulene were the main emitted SQTs. As in 2017 contributions of β-caryophyllene and α-humulene were the highest during early growing season, which indicates that their emissions are related to early growth of the leaves. Later in 2019, β-farnesene was dominating.

### 3.1.3 OSQT emissions

OSQTs were measured only in 2019. High emissions were measured especially during the early growing season (Fig. 2b). Seasonal mean emission rates for OSQT (46–650 ng $g_{dw}^{-1}$ $h^{-1}$) were a bit higher than for SQTs. In total, 9 different OSQTs were detected and the ratios of different OSQTs remained fairly constant over the growing season (Fig. 4d). The only OSQT identified and quantified with an authentic standard was caryophyllene oxide. However, it was <9% of the total measured OSQT mass. In headspace studies of the downy birch leaves by Zang et al. (1999), caryophyllene oxide was also the only OSQT identified and its contribution to the total OSQT mass was 11%. Isidorov et al. (2019) found several different OSQTs in the headspace of downy birch buds and of the OSQTs, 14-hydroxy-β-caryophyllene acetate (M 262 g $mol^{-1}$) had the highest contribution. 14-Hydroxy-β-caryophyllene acetate has also been shown to be the major component of essential oils of birch species native to Turkey (Demirci et al. 2000) and of essential oils of downy and mountain birch buds (Klika et al. 2004). Our mass spectra and retention indexes indicate that OSQT9, the major compound found in our studies, is 14-hydroxy-β-caryophyllene acetate. The second highest OSQT3 was tentatively identified as 6-hydroxy-β-caryophyllene (M 220 g $mol^{-1}$). It was also the second highest contributor of OSQTs in the headspace samples of downy birch buds (Isidorov et al. 2019).

### 3.1.4 Other BVOC emissions

Isoprene emissions were very low (season means < 0.4 ng $g_{dw}^{-1}$ $h^{-1}$) both in 2017 and 2019. The monthly mean emission rates of ALDs, measured only in 2017, were 9 – 23 ng $g_{dw}^{-1}$ $h^{-1}$. Variation of the emission ratios of aldehydes was high (Table A1 in the appendix A). Decanal was the most significant $C_4$-$C_{10}$ aldehyde during the leaf growth, but after that hexanal and nonanal emissions became more important. Hexanal, nonanal and decanal were also major $C_4$-$C_{10}$





aldehydes emitted by a Norway spruce (Hakola et al. 2017). Possanzini et al. (2000) and Bowman et al. (2003) have observed emissions of these larger aldehydes (e.g. heptanal, octanal and nonanal) when ozone attacks the fatty acids on leaf or needle surfaces. However, in our system ozone-free zero air was used.

In 2017 an additional in situ GC-MS2 was used for measuring emissions of GLVs. The time of the measurements did not always overlap with GC-MS1 measurements of terpenes and aldehydes and therefore they cannot unfortunately be directly compared. GLV measurements were conducted during early (26/5 – 7/6/2017), main (21/6 – 19/7/2017) and late (23/8 – 25/8/2017) growing seasons. Seasonal mean emission rates varied between 21 and 240 ng $g_{dw}^{-1}$ $h^{-1}$ (Table A2 in the appendix A). *cis*-3-Hexen-1-ol and *cis*-3-hexenylacetate were the most significant GLVs. Their emissions
were high especially in July (mean 475 ng $g_{dw}^{-1}$ $h^{-1}$, max 8500 ng $g_{dw}^{-1}$ $h^{-1}$), when also some leaf damages were detected. GLVs containing six carbon atoms are emitted directly by plants often as a result of physical damage (Fall, 1999; Hakola et al., 2001). These emissions could also be drought induced since, in 2017, the measured birch was suffering from it. In 2019, the only GLV measured with GC-MS1 was *cis*-3-hexenol, but emissions remained below the detection limit (<30 ng $g_{dw}^{-1}$ $h^{-1}$).

### 3.1.5 The branch chamber temperature

Due to elevated branch chamber temperature compared to ambient conditions, higher than natural emissions may have been induced inside our chamber, especially in sunny conditions. However, this effect is not as high as the difference between ambient air and chamber temperature would indicate since also temperature of the closeby leaves increased
clearly higher than ambient temperatures in sunny conditions (Table 2). Results show that the surface temperature of the leaves next to the chamber in direct sun was on average 8°C higher than ambient air temperature while chamber temperature was 14 °C higher. In partly cloudy conditions difference between leaf surface and ambient air was 7 °C and between chamber and ambient air 10 °C. Without direct sun leaf surface temperatures were 1.4 °C lower than the ambient temperature and chamber temperature was very close to the ambient temperature with the mean difference
being only +0.4 °C.

Table 2. Mean ambient air, downy birch leaf next to the chamber and branch chamber temperatures measured at the same time in the end of July and early August in 2019 in sunny, partly cloudy and cloudy conditions.

| Temperature (°C) | Ambient | Leaf surface (ambient) | Chamber |
|---|---|---|---|
| Sunny | 23.7 | 32.0 | 37.8 |
| Partly cloudy | 26.0 | 32.7 | 35.7 |
| Cloudy | 25.3 | 23.7 | 25.6 |


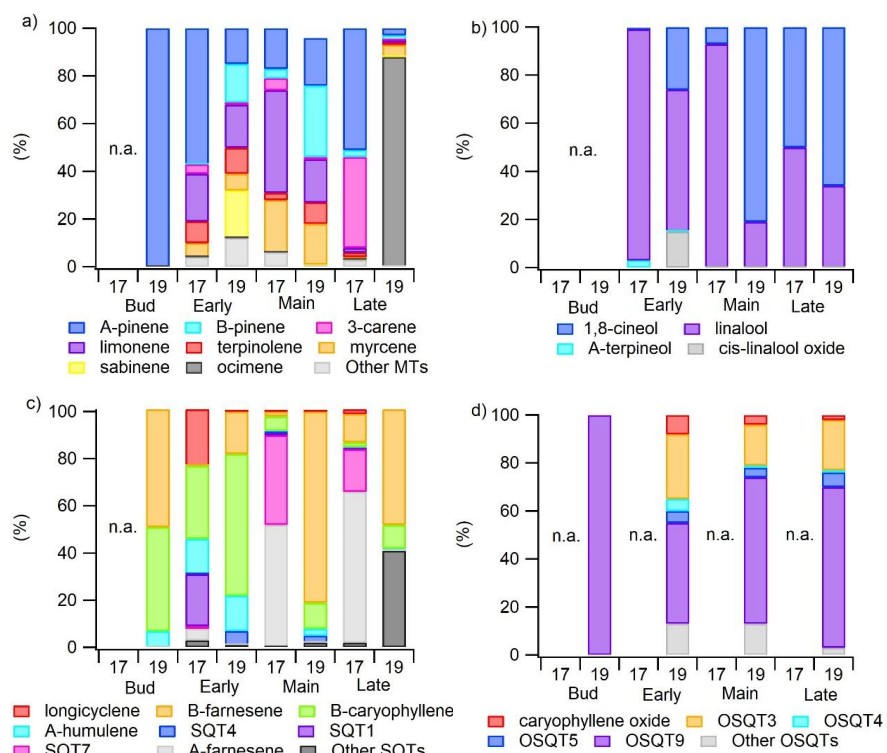

Figure 4: Emission ratios of individual a) MTs b) OMTs, c) SQTs and d) OSQTs during early, main and late growing
seasons in 2017 (17) and 2019 (19).

## 3.2 Variation of the emission ratios between the trees

To study the variability of compounds emitted by different birch trees, ratios of BVOCs were measured from the
headspace of 13 additional birch branches growing close to the main trees in 2019. Of the measured birches, eight were
downy birches, one a silver birch and four were either downy or silver birches. Isoprene had very low contribution in
the emissions of all measured trees over the whole growing season (Fig. 5). As for the main trees measured with the in
situ TD-GC-MS, in the beginning of growing season in May, SQTs and OSQTs were clearly the most important
compounds for most of the trees. Later, contribution of MTs increased and in the end of growing season, the
contributions of SQTs and OSQTs were very low. However, there were clear differences between the trees and for
instance the fraction of MTs varied in May between 3% and 31% (Fig. 5). In their studies of three different downy
birches Hakola et al. (2001) found also highly varying emission ratios of terpenes. For one of their trees MTs and an
OMT (linalool) were clearly the most significant compounds emitted while for the other two trees SQTs had major
contribution. Such inherent diversity in secondary metabolism is often seen in both downy birch and silver birch: Stark
et al (2008) reported qualitative, but not quantitative, latitude-associated gradient in the foliar flavonoids in downy
birch, Makhnev et al. (2012) found genetic differences in triterpene content of silver birch leaf extracts, and Deepak et





al. (2017) associated the genotype with differences in triterpenoids and alcyl coumarates in the surface waxes of silver birch.

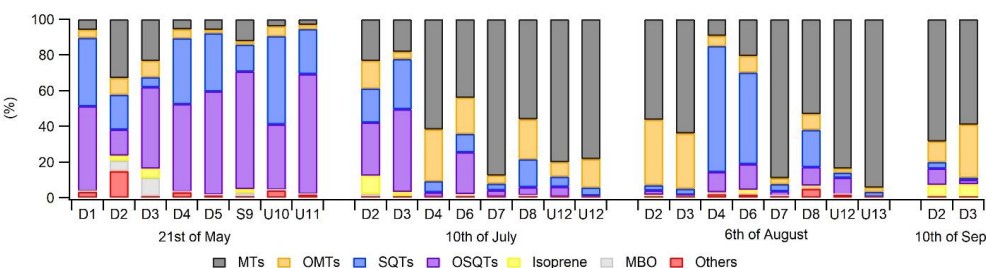

Figure 5: Ratios of BVOC groups in the headspace of different tree branches measured over the growing season in 2019. Numbering indicates different trees, and none were the same as the ones described in Section 2.1. Of the trees D1-D8 were identified as downy birches, S9 was a silver birch and U10-13 were either downy or silver birches.

Ratios of individual terpenes in the head space of the branches can be found from Figure B1 in the appendix B. As for the main trees measured with the in situ TD-GC-MS, of the MTs α-pinene, β-pinene, limonene and sabinene had the highest ratios. However, limonene was detected mainly during the early growing season. In the studies of Isidorov et al. (2019), limonene was the major MT detected in the head space analysis of downy birch buds. Of the OMTs, linalool and one unidentified OMT, had the largest contribution in the emissions of all trees during early growing season, but during the other seasons 1,8-cineol was clearly the major OMT. β-Caryophyllene and β-farnesene were the major SQTs emitted throughout the growing season. One unidentified SQT (SQT4) had occasionally also a strong contribution. β-Caryophyllene and β-farnesene dominated also the emissions of main trees in 2019 (Fig. 4c). Ten different OSQTs were detected in the emissions of studied trees, OSQT9 (identified as 14-hydroxy-β-caryophyllene acetate) being the major emitted compound. Caryophyllene oxide was detected only in the early growing season, while contribution of OSQT4 increased after that. OSQT9 (14-hydroxy-β-caryophyllene acetate) had also major contribution in the emissions from the main tree in 2019 (Fig. 4d).

These results indicate that the tree-to-tree variability in emission patterns seems significant, but still rather small compared to the seasonal variability (growing season). Even though emission ratios vary a lot over the growing season, there seems to be fairly systematic trend in it (i.e. relatively dominant SQT and OSQT ratios during early growing season and increasing contribution of MTS and OMTs during the main and late growing season). However, there can be some birches with very different emission ratios.

**3.3 Temperature and light dependence of the emissions**

Temperature is a significant factor controlling the biogenic emissions (Guenther et al. 2012). However, at least MT emissions from birch leaves are also correlating with light (Hakola et al. 2001, Rinne et al., 2009, Ghirardo et al., 2010). In their $^{13}CO_2$ labelling studies, Ghirardo et al. (2010) found, that all MTs emitted from the silver birch (*Betula*


*pendula*) were from the *de novo* biosynthesis. In darkening experiments of downy birch branches by Hakola et al. (2001), most of the MT emissions declined to values below detection limit very fast after covering the branch, while the darkening effect on SQTs was very low. This indicates that while MTs mainly originate from *de novo* emissions, SQT emissions are likely mostly temperature dependent and coming mainly from the storage pools. However, as shown in Fig. 3, some emissions of MTs were detected also during night especially during high emissions in the early growing season (PAR=0.0 $\mu$mol m$^{-2}$ s$^{-1}$, MT emission 10 ng m$^{-2}$ h$^{-1}$) and therefore they are expected to have some temperature dependent emissions from storage pools as well. We defined both temperature ($E_{30}$) and light and temperature ($E_{CTxCL}$) dependent emission potentials at 30°C and at 1000 $\mu$mol m$^{-2}$ h$^{-1}$ for our downy birch emissions (Table 3).

Table 3. Exponential correlation of emission rates (ng $g_{dw}^{-1}$ h$^{-1}$) with temperature, linear correlation with light and temperature activation factor (CTxCL) and emission potentials during bud break in 2019 and during early, main and late growing seasons in 2017 and 2019. $E_{30}$=temperature dependent emission potential at 30°C, $E_{CTxCL}$= light and temperature dependent emission potential at 30°C and at PAR 1000 $\mu$mol m$^{-2}$ s$^{-1}$, $\beta$=temperature sensitivity, $R^2$=correlation coefficient, a=intercept of the linear fitting of light and temperature dependence curve E=b·CTxCL+a, where E is the emission. Values with correlation coefficient $R^2$<0.3 have been italicized and colored grey.

| 2017 | MTs | | | SQTs | | | ALDs | | |
|---|---|---|---|---|---|---|---|---|---|
| | $E_{30}$ | $\beta$ | $R^2$ | $E_{30}$ | $\beta$ | $R^2$ | $E_{30}$ | $\beta$ | $R^2$ |
| $\underline{T_{30}}$ | ng $g_{dw}^{-1}$ h$^{-1}$ | °C$^{-1}$ | | ng $g_{dw}^{-1}$ h$^{-1}$ | °C$^{-1}$ | | ng $g_{dw}^{-1}$ h$^{-1}$ | °C$^{-1}$ | |
| Early | 319 | 0.06 | 0.43 | 5640 | 0.14 | 0.58 | 350 | 0.15 | 0.61 |
| Main | *4* | *0.03* | *0.04* | 500 | 0.13 | 0.46 | 39 | 0.10 | 0.36 |
| Late | *36* | *0.06* | *0.01* | *10* | *0.02* | *0.00* | 4260* | 0.30 | 0.73 |
| $\underline{CTxCL}$ | $E_{CTxCL}$ | a | $R^2$ | $E_{CTxCL}$ | a | $R^2$ | $E_{CTxCL}$ | a | $R^2$ |
| | ng $g_{dw}^{-1}$ h$^{-1}$ | | | ng $g_{dw}^{-1}$ h$^{-1}$ | | | ng $g_{dw}^{-1}$ h$^{-1}$ | | |
| Early | *270* | *124* | *0.07* | 5100 | 494 | 0.61 | 240 | 37 | 0.57 |
| Main | *5* | *3.8* | *0.02* | 1540 | 50 | 0.40 | 80 | 12 | 0.52 |
| Late | *74* | *-240* | *0.01* | *7* | *11* | *0.00* | 480 | 14 | 0.57 |

| 2019 | MTs | | | SQTs | | | OSQTs | | |
|---|---|---|---|---|---|---|---|---|---|
| | $E_{30}$ | $\beta$ | $R^2$ | $E_{30}$ | $\beta$ | $R^2$ | $E_{30}$ | $\beta$ | $R^2$ |
| $\underline{T_{30}}$ | ng $g_{dw}^{-1}$ h$^{-1}$ | °C$^{-1}$ | | ng $g_{dw}^{-1}$ h$^{-1}$ | °C$^{-1}$ | | ng $g_{dw}^{-1}$ h$^{-1}$ | °C$^{-1}$ | |
| Bud | *44* | *0.01* | *0.14* | 1601 | 0.10 | 0.56 | *543* | *0.03* | *0.11* |
| Early | 187 | 0.15 | 0.70 | 770 | 0.13 | 0.68 | 948 | 0.11 | 0.41 |
| Main | 32 | 0.14 | 0.77 | 74 | 0.14 | 0.57 | 340 | 0.07 | 0.33 |
| Late | *63* | *0.06* | *0.07* | *85* | *0.08* | *0.26* | 416 | 0.09 | 0.31 |
| $\underline{CTxCL}$ | $E_{CTxCL}$ | a | $R^2$ | $E_{CTxCL}$ | a | $R^2$ | $E_{CTxCL}$ | a | $R^2$ |
| | ng $g_{dw}^{-1}$ h$^{-1}$ | | | ng $g_{dw}^{-1}$ h$^{-1}$ | | | ng $g_{dw}^{-1}$ h$^{-1}$ | | |
| Bud | n.a. | n.a. | n.a. | n.a. | n.a. | n.a. | n.a. | n.a. | n.a. |





| Early | 370 | 20 | 0.62 | 1590 | 108 | 0.33 | *2317* | *1858* | *0.15* |
| Main | 76 | 2 | 0.72 | 189 | 7 | 0.55 | *429* | *253* | *0.11* |
| Late | *405* | *99* | *0.13* | *149* | *101* | *0.03* | *572* | *315* | *0.08* |

*Due to emission rates close to the detection limits and low chamber temperature (<18.7°C) upscaling to the 30°C can give unrealistic value. Linear fitting gives as high $R^2$ value, but then the emission potential is only 124 ng $g_{dw}^{-1}$ $h^{-1}$.

Emission potentials showed a very high variation between the seasons (Table 3). The highest potentials were measured during the bud break and early growing season.  Even if the bud break emission potentials are high especially for the SQTs, real emissions are still low since the leaf mass is very low (<10% of the mass of fully grown leaves) during that period. After the early growing season emission potentials decreased considerably in June and July. During the late growing season emissions of SQTs and OSQTs were as low as during the main growing season, but for MTs relatively

high emissions were detected during the last two days of measurements in 2019.

Of the OSQTs the only compound detected during the bud break period was OSQT9 (14-hydroxy-β-caryophyllene acetate). Correlation with temperature during that time was low ($R^2$=0.11, Table 3). If we use a 2h time lag for temperature, correlation was clearly higher ($R^2$=0.39) and emission potential of OSQT9 was 615 ng $g_{dw}^{-1}$ $h^{-1}$. However,

even with this high emission potential, emissions are expected to be low due to the low biomass of the leaves during bud break. Also, during the early growing season correlation with temperature was better while using the temperature measured two hours earlier. It is possibly explained by delayed temperature response of these less volatile compounds in the leaves and losses and re-evaporation of them on the inlet and branch chamber walls. In our earlier tests we have detected some losses of caryophyllene oxide into our inlet system and particularly in the instrument (Helin et al. 2020).

Based on this and due the fact that OSQT9 had higher molecular weight, more oxygen atoms and more double bonds than caryophyllene oxide, it is possible that the real emissions may be even higher than those observed here.

For the ALDs, very high emission potential (4260 ng $g_{dw}^{-1}$ $h^{-1}$) was observed during the late growing season in August 2017 (Table 3). Even though the linear fit of the temperature dependence resulted in high $R^2$ value, the emission

potential was only 124 ng $g_{dw}^{-1}$ $h^{-1}$ when using the linear fitting ($R^2$=0.73) for the temperature dependence curve. Chamber temperature during these late growing season measurements was always <18.7 °C and most of the measured emission rates were very close to the detection limits. The highest measured emission rate of aldehydes was 103 ng $g_{dw}^{-1}$ $h^{-1}$. Therefore, the upscaling of these results to  30 °C may be unrealistic.

GLVs, measured only in 2017, were slightly correlated with temperature during the main ($R^2$=0.41) and late ($R^2$=0.31) growing seasons, but temperature was not the main factor controlling their emission. GLV emissions coincided with leaf damages and were potentially stress induced (Section 3.4).

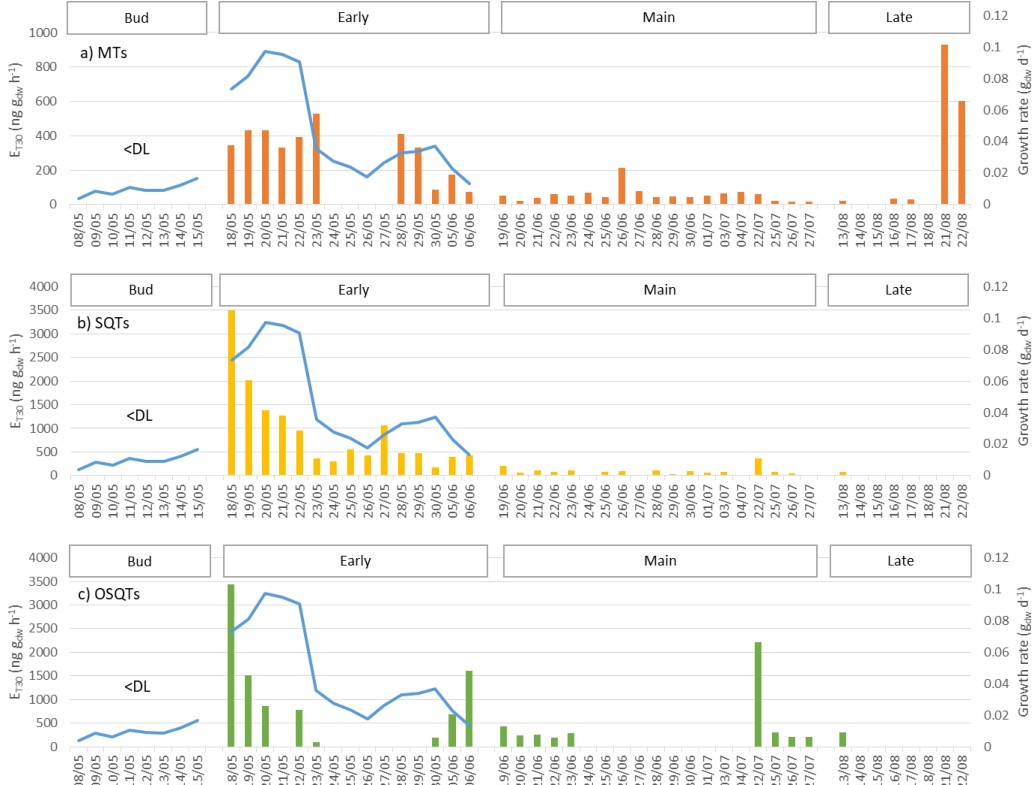

Figure 6: Daily mean emission potentials at 30°C ($E_{30,day}$) of a) MTs, b) SQTs and c) OSQTs (bars) during bud break, early, main and late growing seasons in 2019 and modeled growth rate of leaves ($g_{dw}$ day$^{-1}$) of the measured branch (blue line). During the budbreak period (<DL) emissions over the day were below detection limits and therefore the calculation of daily emission potential was not possible. Emission potentials of days with low temperature correlation ($R^2 < 0.3$) were omitted. Potentials were calculated only for the days with more than 12h measured.

Daily mean emission potentials ($E_{30,day}$) were calculated for 2019 for all days with more than half of the measurements available. During the early growing season, the mean daily correlation coefficients ($R^2$) between the emission rates and temperature were 0.84, 0.86 and 0.51 for MTs, SQTs and OSQTs, respectively. Even though short term and diurnal variations of the emissions were explained well by the temperature, the development of the buds and growth of leaves had a very strong effect on the emission potentials (Fig. 6). The highest daily emission potentials were measured during the highest growth rate and a decrease followed with the decrease of the growth rate of the leaves. This suggests that there could be storage pools of these compounds in the buds, which are released after the bud break. Isidorov et al. (2019) detected huge amounts of different VOCs, including SQTs and OSQTs, in the headspace of downy birch buds. For OSQTs additional peak in the emission potentials was observed on 22 July.





For MTs, the highest daily mean emission potentials were detected during the last two days of measurements in August. As mentioned before, this was due to the high emissions of one MT, tentatively identified as β-ocimene. It is possible that ocimene emissions are related to the senescence of leaves, but also herbivore induced ocimene emissions have been detected (Vuorinen et al. 2007).

Emission potentials found in earlier studies are listed in Table 4. Even though at least MT emissions from birches are known to be also light dependent (Hakola et al. 2001, Rinne et al., 2009, Ghirardo et al., 2010), most studies report emission rates normalized to only by temperature. Therefore, only temperature ($E_{30}$) normalized emission potentials are compared here. Emission potentials reported vary over 3 orders of magnitude, the lowest MT emission potential of

10 downy birch being only 5 ng $g_{dw}$ h$^{-1}$ and the highest 5000 ng $g_{dw}$ h$^{-1}$. For downy birch SQT emissions the lowest reported emission potential was 17 ng $g_{dw}$ h$^{-1}$ and the highest 6900 ng $g_{dw}$ h$^{-1}$. It is noteworthy that most of the earlier studies are only from short periods or laboratory conditions, and do not report the seasonal variation of the emission potentials. As shown here for MTs, SQTs and OSQTs and by Hakola et al. (2001) for MTs, seasonal variation is very high and this could explain at least part of the differences in results from literature. Variation in MT and SQT emission

15 potentials between birch species is up to several magnitudes when comparisons have been made in similar conditions.

Table 4. Comparison of emission potentials normalized to temperature 30°C ($E_{30}$, ng $g_{dw}^{-1}$ h$^{-1}$). Emission potentials with very low correlation ($R^2<0.3$) has been omitted from this study data. ETS=effective temperature sum describing the state of the growing season.

| Birch type | Season | MTs ($E_{30}$) | SQTs ($E_{30}$) | OSQTs ($E_{30}$) | Refs. |
|---|---|---|---|---|---|
| Downy | Early | - | 5110 | - | This study, 2017 |
| | Early | 190 | 770 | 950 | This study, 2019 |
| | ETS<80 | 1470 | - | - | Hakola et al. (2001), tree 1 |
| | Main | - | 500 | - | This study (2017), stress |
| | Main | 32 | 74 | 340 | This study (2019) |
| | 80<ETS<400 | 720 | - | - | Hakola et al. (2001), tree 1 |
| | 80<ETS<400 | 310 | - | - | Hakola et al. (2001), tree 2 |
| | 80<ETS<400 | 310 | - | - | Hakola et al. (2001), tree 3 |
| | Late | 5 | - | - | This study, 2017 |
| | Late | - | - | 420 | This study, 2019 |
| | ETS>400 | 5490 | - | - | Hakola et al. (2001), tree 1 |
| | ETS>400 | 1710 | - | - | Hakola et al. (2001), tree 2 |
| | ETS>400 | 170 | - | - | Hakola et al. (2001), tree 3 |
| | Summer | - | 310 | - | Hakola et al. (2001), tree 1 |
| | Summer | - | 6940 | - | Hakola et al. (2001), tree 2 |
| | Summer | - | 810 | - | Hakola et al. (2001), tree 3 |
| | unspecified | 3000 | 2000 | - | Karl et al. (2009) |



| | | | | |
|---|---|---|---|---|
| | laboratory | 5-10 | 17-31 | - | Räsänen et al. (2017)[a] |
| Silver | ETS<80 | 3630 | - | - | Hakola et al. (2001) |
| | 80<ETS<400 | 680 | - | - | Hakola et al. (2001) |
| | ETS>400 | 7710 | - | - | Hakola et al. (2001) |
| | unspecified | 3000 | 2000 | - | Karl et al. (2009) |
| | laboratory | 23-188 | 41-190 | - | Räsänen et al. (2017)[a] |
| Mountain | Main | 31 | 35 | -- | Ahlberg et al. (2011) |
| | Main | 5300 | 6500 | - | Haapanala et al. (2009) |

[a] Converted using foliar density 240 g m$^{-2}$ obtained from Karl et al. (2009)

### 3.4 The effect of stress on the emissions

In July 2017 the branch was visibly suffering from leaf damages (Fig. 1) and possibly effects of drought and increased chamber temperature (>30ºC). During that time very high emissions of two GLVs, *cis*-3-hexen-1-ol and *cis*-2-hexenylacetate, were detected (max. 1500 and 6900 ng $g_{dw}^{-1}$ h$^{-1}$, respectively). In July simultaneously with GLV emissions, high emissions of two SQTs (SQT7 and α-farnesene) were detected as well (max. 4700 and 1800 ng $g_{dw}^{-1}$ h$^{-1}$, respectively). However, emissions of GLVs were the highest in the afternoon on 18 July, while for the SQTs the highest emissions were measured on 16 July. Relatively high α-farnesene emissions were detected also in the end of June together with emission of *cis*-3-hexen-1-ol and *cis*-2-hexenylacetate, but SQT7 remained below detection limits during that time. Also some linalool emissions (up to 23 ng $g_{dw}^{-1}$ h$^{-1}$) were detected during these stress periods. In 2019, emissions of all these compounds remained very low or below detection limits.

While SQT7 emission potentials during the stress period in July 2017 had very high correlation with the chamber temperature ($R^2$=0.95, β=0.17 ºC$^{-1}$, E$_{30}$=268 ng $g_{dw}^{-1}$ h$^{-1}$), and also α-farnesene showed some temperature dependence ($R^2$=0.53, β=0.11 ºC$^{-1}$, E$_{30}$=209 ng $g_{dw}^{-1}$ h$^{-1}$), *cis*-3-hexen-1-ol and *cis*-2-hexenylacetate emissions did not follow the changes in temperature or light ($R^2$=0.21 and 0.31, respectively) but were the highest on 18 July when the temperature was lower than the previous days.

Results indicate that even though *cis*-3-hexen-1-ol, *cis*-2-hexenylacetate, α-farnesene, linalool and SQT7 emissions seemed to be stress related, mechanisms behind their emissions may be different. Earlier studies have found that GLVs are emitted by plants as a result of physical damage (Fall, 1999; Hakola et al., 2001) and α-farnesene emissions are known to be highly sensitive to the biotic stress (Kännaste et al. 2008, Faiola and Taipale, 2020).

### 3.5 Atmospheric implications of the emissions

In their emission inventory for the boreal ecosystem in Finland, Tarvainen et al. (2007) estimated that MTs are the major terpene group emitted (84% total flux) and that SQTs are mostly only emitted in July and August, with downy and silver birches as the largest contributor. The share of SQTs on total terpene flux was estimated to be 7%. Taking into account our results with high emission potentials of SQTs already in May and June, the importance of SQTs would be much higher. The oxidation products of VOCs play a key role in secondary aerosol (SOA) formation, especially in forested areas (Ehn et al., 2014; Kulmala et al., 2013) and SQTs produce higher comparative aerosol yields than


isoprene and abundant MTs, and consequently may contribute significantly to SOA formation (Griffin et al. 1999; Lee et al., 2006, Frosch et al. 2013). Oxidation products of SQTs have very high cloud condensation activity and therefore these birch emissions potentially have high impacts also on cloud formation (Be et al., 2017). OSQT emission potentials were even higher than for SQTs in our study. With their higher molecular weights and lower volatilities they

could have even higher SOA yields than SQTs, but to our knowledge, this has not been studied yet. Due to very high reaction rates of SQTs and possibly OSQTs with $O_3$ and hydroxyl radicals (OH), their effects on local oxidation capacity is also expected to be strong and should warrant further investigation.

**4 Conclusions**

Even though isoprene is the main BVOC emitted globally (Gunther et al. 1995), it is clear that downy birches do not contribute to its emissions. The main VOC groups emitted by the downy birches were SQTs and OSQTs especially in the beginning of the boreal summer in May and early June. Downy birches were also a source of MTs, ALDs and GLVs. However, emission ratios of the studied VOCs were highly variable over the growing season.

The highest BVOC emissions were detected during the early growing season indicating that early growth of leaves is a strong source of these compounds. Of the SQTs, especially β-caryophyllene and α-humulene emissions were clearly related to early growth season, whereas, later, their emissions were very low or below detection limits. Of the OSQTs, two compounds tentatively identified as 14-hydroxy-β-caryophyllene acetate and 6-hydroxy-β-caryophyllene had a major contribution over the whole growing season with emissions being clearly the highest during the early growing

season. In 2017, leaf damages and stress possibly related to drought and high chamber temperature was found to induce emissions of GLVs (*cis*-3-hexen-1-ol and *cis*-2-hexenylacetate), linalool, α-farnesene and an unidentified SQT. The emissions of BVOCs were peaking in the afternoon together with temperature and PAR and were very low during the night.

Earlier studies have shown that MT emissions of downy birches are mainly light dependent, and our results mainly agree with this although some nighttime emissions were also detected. However, our results indicated that emissions of SQTs and OSQTs from downy birch foliage are mostly temperature dependent. The highest seasonal emission potentials of MTs, SQTs and OSQTs were observed during the bud break and early growing season. Daily emission potentials were the highest during the fastest leaf growth period and a decrease in emission potentials followed the

decrease of the growth rate of the leaves. This indicates that there could be storage pools of these compounds in the buds, which are released after the bud break. Due to high variability of the emissions over the growing season, it is clear that estimating birch emissions should take into account the seasonality of emission potentials.

The results indicate that downy birch can be a significant contributor for the relatively high SQT concentrations found

in the boreal forest air (Hellén et al., 2018). These emissions may have strong effects especially on SOA formation. In addition, emissions of OSQTs are as high as SQT emissions and may have strong impacts in the atmosphere as well. SOA formation potential of downy birch emissions is expected be high especially during the early growing season due to high emissions of compounds with high SOA formation potentials (SQTs, OSQTs and limonene).



**Author contributions**

H. Hellén designed and conducted the VOC measurements, performed the data analysis and led the writing of the manuscript. H. Hakola supervised the study, helped designing the measurement campaign and the commented on the manuscript. S.Schallhart, T.Tykkä and A. Praplan conducted the VOC measurements and data analysis and commented

on the manuscript. P.P. Schiestl-Aalto and J. Bäck provided the trees to measure in 2017 and data on leaf growth and wrote their description and commented on the manuscript.

**Acknowledgements**

The research was supported by the Academy research fellow projects (Academy of Finland, project No. 275608 and No. 307797), Academy of Finland project (No. 316151), Knut and Alice Wallenberg Foundation (no. 2015.0047) and

Academy of Finland via the Center of Excellence in Atmospheric Sciences (No. 307331).

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





**Appendix A: Emission rates of the measured compounds**

Table A1. Emission rates (ng $g_{dw}^{-1}$ $h^{-1}$) of downy birches measured with GC-MS1 together with temperature (T),
relative humidity (RH) and photosynthetic radiation (PAR) on the branch chamber during the measurements in
2017 and 2019. N= number of samples.

| ng $g_{dw}^{-1}$ $h^{-1}$ | 2017 | | | 2019 | | | |
|---|---|---|---|---|---|---|---|
| | Early | Main | Late | Bud | Early | Main | Late |
| N | 43 | 249 | 114 | 179 | 319 | 434 | 209 |
| T (ºC) | 12 | 14 | 10 | 9 | 16 | 19 | 17 |
| RH (%) | 19 | 28 | 43 | 44 | 82 | 66 | 63 |
| PAR (µmol $m^{-2}$ $s^{-1}$) | 186 | 144 | 68 | n.a. | 220 | 164 | 131 |
| isoprene | 0 | 0 | 0 | 0 | 0 | 0 | 0 |
| 2-methyl-3-buten-1-ol | 0 | 0 | 0 | 0 | 1 | 1 | 0 |
| cis-3-hexenol | n.a. | n.a. | n.a. | 0 | 0 | 0 | 6 |
| α-pinene | 58 | 1 | 32 | 2 | 15 | 3 | 2 |
| camphene | 0 | 0 | 1 | 0 | 0 | 0 | 0 |
| β-pinene | 0 | 0 | 2 | 0 | 16 | 5 | 1 |
| 3Δ-carene | 4 | 0 | 24 | 0 | 1 | 0 | 0 |
| p-cymene | 2 | 0 | 0 | 0 | 1 | 0 | 0 |
| limonene | 20 | 2 | 1 | 0 | 17 | 3 | 1 |
| terpinolene | 9 | 0 | 2 | 0 | 10 | 2 | 1 |
| myrcene | 6 | 1 | 1 | 0 | 6 | 2 | 3 |
| sabinene* | n.a. | n.a. | n.a. | 0 | 19 | 3 | 0 |
| β-phellandrene* | 2 | 0 | 1 | 0 | n.a. | n.a. | n.a. |
| 4Δ-Carene* | | n.a. | n.a. | 0 | 5 | 0 | 0 |
| β-ocimene* | n.a. | n.a. | n.a. | 0 | 0 | 0 | 57 |
| **MTs** | **129** | **15** | **64** | **2** | **88** | **17** | **64** |
| 1,8-cineol | 0 | 1 | 0 | 0 | 10 | 3 | 0 |
| linalool | 28 | 10 | 0 | 0 | 23 | 1 | 0 |
| α-terpineol | 1 | 0 | 0 | n.a. | n.a. | n.a. | n.a. |
| cis-linalool oxide | n.d | n.a. | n.a. | 0 | 6 | 0 | 0 |
| **OMTs** | **29** | **11** | **0** | **0** | **39** | **4** | **1** |
| longicyclene | 161 | 0 | 0 | 0 | 1 | 0 | 0 |
| iso-longifolene | 0 | 0 | 0 | 0 | 0 | 0 | 0 |
| β-farnesene | 2 | 4 | 1 | 53 | 91 | 33 | 7 |
| β-caryophyllene | 209 | 14 | 0 | 47 | 297 | 4 | 1 |
| α-humulene | 101 | 2 | 0 | 7 | 77 | 1 | 0 |
| α-farnesene | 37 | 117 | 3 | 0 | n.a. | n.a. | n.a. |
| SQT1 | 150 | 2 | 0 | 0 | 0 | 0 | 0 |
| SQT2 | 3 | 0 | 0 | 0 | 3 | 0 | 0 |
| SQT3 | 5 | 0 | 0 | 0 | 1 | 0 | 0 |
| SQT4 | 1 | 0 | 0 | 0 | 29 | 1 | 0 |
| SQT5 | 11 | 0 | 0 | 0 | 2 | 1 | 1 |
| SQT6 | 1 | 0 | 0 | 0 | 0 | 0 | 5 |
| SQT7 | 9 | 86 | 1 | 0 | n.a. | n.a. | n.a. |





| SQTs | 692 | 226 | 5 | 108 | 505 | 41 | 14 |
|---|---|---|---|---|---|---|---|
| caryophyllene oxide | n.a. | n.a. | n.a. | 0 | 54 | 5 | 1 |
| OSQT1 | n.a. | n.a. | n.a. | 0 | 13 | 3 | 0 |
| OSQT2 | n.a. | n.a. | n.a. | 0 | 20 | 0 | 0 |
| 6-hydroxy-β-caryophyllene* | n.a. | n.a. | n.a. | 0 | 174 | 22 | 10 |
| OSQT4 | n.a. | n.a. | n.a. | 0 | 30 | 2 | 1 |
| OSQT5 | n.a. | n.a. | n.a. | 0 | 32 | 5 | 3 |
| OSQT6 | n.a. | n.a. | n.a. | 0 | 29 | 6 | 1 |
| OSQT7 | n.a. | n.a. | n.a. | 0 | 3 | 0 | 0 |
| OSQT8 | n.a. | n.a. | n.a. | 0 | 18 | 8 | 0 |
| 14-hydroxy-β-caryophyllene acetate* | n.a. | n.a. | n.a. | 47 | 272 | 82 | 31 |
| **OSQTs** | **n.a.** | **n.a.** | **n.a.** | **47** | **651** | **134** | **46** |
| pentanal | 1 | 1 | 1 | n.a. | n.a. | n.a. | n.a. |
| hexanal | 5 | 4 | 6 | n.a. | n.a. | n.a. | n.a. |
| heptanal | 2 | 1 | 2 | n.a. | n.a. | n.a. | n.a. |
| octanal | 1 | 1 | 3 | n.a. | n.a. | n.a. | n.a. |
| nonanal | 0 | 6 | 10 | n.a. | n.a. | n.a. | n.a. |
| decanal | 23 | 3 | 2 | n.a. | n.a. | n.a. | n.a. |
| **ALDs** | **33** | **16** | **23** | **n.a.** | **n.a.** | **n.a.** | **n.a.** |

* Tentatively identified, n.a.=not available

Table A2. Emission rates (ng $g_{dw}^{-1}$ $h^{-1}$) of downy birches measured with GC-MS2 together with temperature (T), relative humidity (RH) and photosynthetic radiation (PAR) in the branch chamber during the measurements in 2017. N= number of samples.

| ng $g_{dw}^{-1}$ $h^{-1}$ | Early | Main | Late |
|---|---|---|---|
| N | 136 | 158 | 25 |
| T (ºC) | 11 | 14 | 11 |
| RH (%) | 27 | 28 | 49 |
| PAR (μmol $m^{-2}$ $s^{-1}$) | 199 | 141 | 62 |
| 1-hexanol | 1 | 1 | 2 |
| 1-octen-3-ol | 0 | 0 | 0 |
| 1-pentanol | 0 | 3 | 0 |
| hexylacetate | 0 | 0 | 0 |
| *cis*-2-hexen-1-ol | 0 | 0 | 0 |
| *cis*-3-hexen-1-ol | 1 | 39 | 55 |
| *cis*-3-hexenylacetate | 20 | 195 | 84 |
| *trans*-2-Hexen-1-ol | 0 | 5 | 3 |
| *trans*-2-hexenylacetate | 0 | 2 | 0 |
| *trans*-3-hexen-1-ol | 0 | 0 | 0 |
| **GLVs** | **22** | **243** | **144** |

**Appendix B: Additional birch branches**

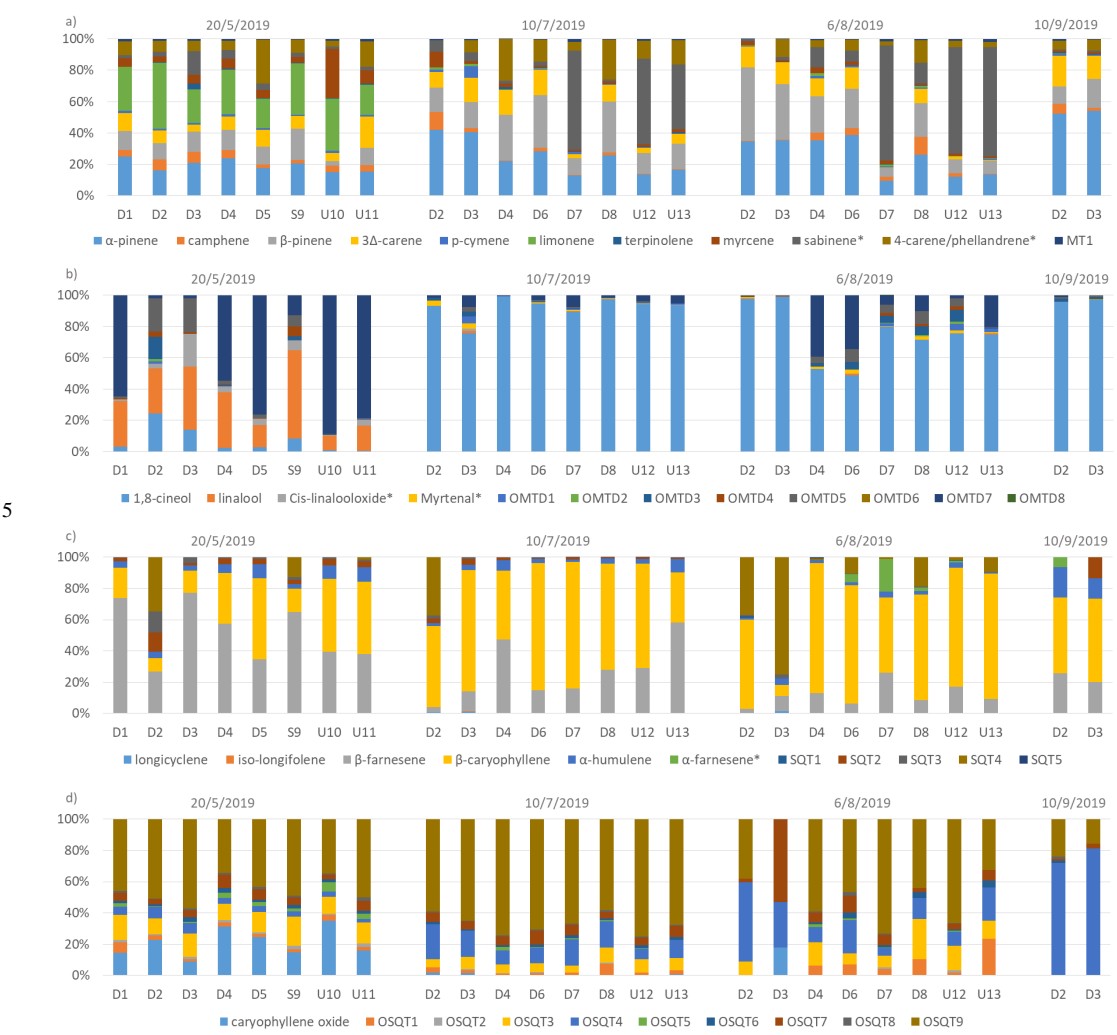

Figure B1. Ratios of a) MTs, b) OMTs, c) SQTs and d) OSQTs in the head space of 13 different birch branches
measured over the growing season in 2019. Of the trees D1-D8 were identified as downy birches, S9 was a
silver birch and U10-13 were either downy or silver birches.