# Peer review of "Sesquiterpenes and oxygenated sesquiterpenes dominate the VOC $(C_5-C_{20})$ emissions of downy birches"

_Atmospheric Chemistry and Physics, 2020_

## Referee Comment (RC1) · Anonymous Referee #1 · 10 Jan 2021

This manuscript provides a large and useful dataset describing seasonal variation in BVOC emissions from Betula pubescens, the most important deciduous tree species of the Eurasian boreal forest. Of particular significance is the inclusion of a number of previously understudied compounds, sesquiterpenes and oxygenated sesquiterpenes, which are expected to play an oversized role in atmospheric chemistry due to their potentially high rates of reactivity and SOA formation. Inclusion of such reactive compounds may help to reconcile discrepancies between leaf level estimates of OH reactivity and measurements of OH reactivity in the forest atmosphere. The authors have made a large number of repeated measurements on single branches across each of two growing seasons and supplemented their data set with qualitative BVOC emissions characterizations of branches of an additional 13 trees. As a consequence, they ob-

served a great amount of variability across seasons and between tree individuals, both quantitatively and qualitatively, i.e., the chemical species composition of emissions. This high amount of variability necessarily results in a messy, somewhat confusing dataset and complicates data interpretation, but documenting the variability is itself an important result. And despite this variability, the authors were able to draw fairly robust general conclusions about seasonal trends and the relative importance of different classes of emitted BVOC over time.

While I regard these emission rate data and the conclusions drawn as qualitatively valid and worthy of publication, I have a number of reservations regarding the quantitative validity of their measurements and their calculated BVOC emission factors, as outlined below.

Apparently, and inevitably when a singe branch is measured repeatedly over an entire season, leaf biomass could not be measured until the end of the experiment. Therefore, biomass was necessarily estimated, in this case using a growth model developed for needles of Scots pine. Its validity for deciduous species is unaddressed. While estimates for mature leaves, when growth has slowed or stopped are likely to be quite accurate, biomass estimates during bud break and early leaf growth seem highly problematic, and errors could lead to corresponding errors in emission estimates. Given the observed high emission rates of SQT and OSQT immediately following bud break, one might wonder if underestimates of leaf biomass contribute to these high rates. Was there any effort to determine early season leaf biomass, specific leaf mass, etc. on comparable nearby branches or to validate the CASSIA model for B. pubescens?

Branch enclosures lacking any sort of temperature control or within chamber mixing, relying solely on air flow through the chamber, inevitably experience above ambient temperatures under sunlit or partial sun conditions. The authors address this issue, characterizing the difference between ambient temperatures inside and outside the enclosure, but the overtemperatures in the chamber are enormous, averaging 10 to 14 deg. under partly cloudy and sunny conditions, respectively. Given typical Q10 values

for BVOC emissions, this would likely result in 2 to 4-fold increases in branch emissions within the chamber compared to those outside the chamber. Thus, reported emission rates should be viewed with some skepticism.

Of course, the relevant temperature for characterizing emissions is the leaf temperature, rather than chamber air temperature. Using an infrared thermometer, the authors characterize the relationship between air temperature and leaf temperature on branches outside the chamber. As expected, these data suggest that sunlit leaves outside the measurement enclosure experience temperatures significantly above ambient, as much as 8 deg. in sunny conditions (which seems a little high). The authors use this data to suggest that, since leaves outside the chamber experience temperatures significantly above ambient air temperature, the elevated air temperature within the enclosure is not so problematic. This ignores the possibility, however, that leaf temperatures within the enclosure are also significantly elevated above chamber ambient air temperatures under partial or full sun conditions, further exacerbating the overtemperature problem. While the authors maintain (p. 5, l. 22) that a flow rate of 3.0 lpm through a 6 liter enclosure was sufficient "to keep the leaf temperature close to the chamber air temperature" they offer no evidence in support of that statement. One wishes the authors had placed thermocouples on leaves inside the chamber, i.e., leaves actually measured, ideally throughout the course of the experiment, but at the very least in order to validate their assumptions regarding within chamber leaf temperatures.

In a related issue, the use of PAR, measured above the enclosure, is of limited utility when trying to characterize the actual irradiation on a multitude of leaves in a branch enclosure, where leaf angle and self-shading can drastically reduce the light reaching an individual leaf. For a certainty, the average PAR received by leaves within the enclosure is significantly less than that measured above the chamber. Since at least some of the BVOC emissions result from de novo production, dependent on light, inadequate characterization of PAR represents a problem, particularly when trying to apply the Guenther et al. light algorithm to estimate emission potentials, discussed further

below.

With respect to de novo emissions versus emissions from storage pools, the analysis presented here is somewhat confusing. Citing previous studies employing 13C to identify de novo production of monoterpenes in birch species or studies where emissions fell to zero upon chamber darkening, the authors quite reasonably assume that some or all of the monoterpenes measured in this study would evidence a light dependency. If one assumes that most or all of MT emissions arise de novo, why solve for solve for emission potential using only a temperature function (Table 3)? A simple darkening of their enclosures would have removed any doubt, as well as providing information regarding light dependency of other classes of BVOC, such as SQT and OSQT, some of which (including B-caryophyllene, B-farnesene and linalool, all significant emissions in this study) are at least partially the result of de novo production in other tree species (e.g., Ponderosa pine (Harley et al. 2014), although Hakola et al. (2001) report little effect of darkening on SQT emissions in B. pubescens. Although the authors assume that the observed SQT emissions arise from storage pools, and are therefore dependent on temperature alone, they nevertheless present data in Table 3, calculating SQT (and OSQT and ALD) emission potentials assuming both light and temperature dependencies. If one examines Fig. 3, the lack of a light dependency for SQT and OSQT emissions is indeed called into question. If one compares the early season data for MT, SQT and OSQT emissions, the shape of the responses is almost identical. That is, all show very low (but not zero) nighttime emissions, followed by a more than 10-fold increase during the day. This large an increase is very unlikely to be explained simply by a 17 deg temperature change, requiring a $Q_{10}$ of over 5. With respect to MT, this large increase is easily explained by including a light response. But what about SQT and OSQT? Again, a simple darkening of the enclosure would have helped resolve the issue.

Throughout the MS, seasonal means of emissions are reported. If I understand correctly, these are simply the means of all measurements made during a given season.

This has some information value. But since the majority of measurements were apparently made during periods of darkness or low light and low nighttime temperatures (using data points in Fig. 3 as a guide), seasonal means significantly underestimate mean midday maximal emissions. For example, the early seasonal mean of all SQT emissions is 692, while the midday maximum in Fig. 3 is well over 2000. Using seasonal means of all data seriously underestimates the significance of midday emissions, of greatest importance for most atmospheric chemistry issues. Nowhere in the MS are mean daily maxima of emissions reported. Similarly, reporting only seasonal means of PAR (including nighttime PAR) and temperature is of modest values; reporting daytime means or maxima would be of more use.

Given the issues related to leaf temperature and PAR incident on leaves, I have very little confidence in any of the emission potential estimates provided in the manuscript. In addition to the temperature and light issues, I have some issues with the general procedures used to determine emission potentials. If I understand correctly (the authors don't present much detai) all of the data in a given season are lumped together and the data are fit to either the Guenther temperature algorithm or the light and temperature algorithms. I am unsure how appropriate, or useful, this is, particularly if the majority of the data is obtained under conditions of darkness and relatively low temperature. If I've misunderstood how these emission potentials are arrived at, I apologize, but more detail about the procedure might be warranted. Finally, given the considerable variability in emissions even within a given season (Fig. 6), I question the utility of publishing seasonal mean values. The utility of calculating these emission potentials by the methods employed is further called into question by Table 3, in which almost half of the determinations are not considered sufficiently robust to report, while others generate unrealistic estimates of Beta.

Thus, I recommend that the authors focus more on the measured emission rates and less on the attempts to determine emission capacities. They should present their emission data, being straightforward about the enclosure temperature issues and the resulting impact on their measurements. Having done so, they can still draw reasonable conclusions regarding the suite of BVOCs emitted by B. pubescens, the seasonal and tree to tree variability, and about the importance of previously unreported OVOC or OSQT emissions.

A few other issues require some attention, as follows.

Abstract, line 24. See comments above regarding the reporting of seasonal means).

Fig. 1. Early dates in 2017 should be 24-28/5 (although I find 24-28 May preferable)

Reconcile p. 5, line 12 in which losses are "negligible" with p. 15, line 19 in which significant (?) losses are implied for high MW OSQTs.

p. 2, line 35. "These OSQTs are expected to be highly reactive and to have higher comparative secondary organic aerosols yields. . ." Likely true, but please provide a reference.

p. 4, line 15. How long was the branch enclosed in the chamber during measurements? How soon after enclosing the branch was sample collection begun?

p. 5, line 9. What was the sampling duration? I.e., how large a sample collected?

p. 5, line 22. I'm not convinced that a flow rate of 3 lpm through a 6 liter enclosure is sufficient to ensure that leaf temperatures remain close to enclosure temperature. Do the authors have evidence? Measuring temperature of leaves outside the enclosure does not adequately characterize leaf temperatures within the chamber. Why were leaf temperatures not measured on leaves inside the enclosure, ideally on a continuous basis, but certainly for brief periods to test these assumptions under varying levels of irradiance?

p. 6, line 11. Empty chamber blanks were subtracted. Please give some idea of the magnitude of these corrections. Can they be considered negligible?

p. 7, line 16. Since you are "solving" for E30, shouldn't the form of the equation be:

[Figure]

E30 = E/(exp(Beta*(T-Ts)))

Of more significance, what value of Beta was used, or were Beta and E30 solved for simultaneously? It appears from Table 3 that they were solved for simultaneously, with values of Beta ranging from 0.03 to 0.15. Guenther 93 reported a range of Beta values for different species varying from 0.057 to 0.144 and uses a mean value of Beta=0.09. I see no reason why the temperature dependency of emissions for a single species should vary so widely, and in the absence of experimental determination of Beta (emission measurements while varying leaf temperature under otherwise constant conditions), I think choosing a constant value for Beta would be the wiser course.

p. 8, line 20. "This was the case particularly in SQT and OSQT emissions." It's not clear to me what this sentence refers to. Are you implying that SQT and OSQT are primarily released from storage pools? Based on what evidence? The daily pattern of emissions of MT, SQT and OSQT in Fig. 3 are almost indistinguishable (discussed further above).

Fig. 3. Please indicate whether these are plots of individual days within each season or plots of seasonal means? If individual days, which of the days shown in Fig. 6 (in which emission potentials varied widely within a season) was chosen? Is it representative, i.e. typical daily behavior?

p. 10, lines 3 and 22. As I've indicated, I question the value of reporting mean emission rates by season. However, if you choose to do so, saying that "seasonal mean emission rates were 5 – 690" conveys very little information and is actually confusing. Something like "Mean emission rates in 2017 were significantly higher in May (692) than in June-July (226) or August (5). Similarly, in 2019. . ."

p. 11, line 8. As discussed, GLV emissions are clearly associated with leaf damage/stress. Given that, seasonal mean values are more or less meaningless. I suggest eliminating Table A2 and stressing the highest values of GLV emissions as representing stress or damage responses.

p. 13, line 19 and elsewhere. Was OSQT9 identified as 14-hydroxy-$\beta$-caryophyllene acetate or not? If so, refer to it by name; if not, say "tentatively identified as . . ."

p. 15, line 5 "Emission potentials showed a very high variation between the seasons." This is true, but emission potentials also showed large variation within seasons (Fig. 6).

Fig. 6. Are these emission potentials (ET30 or E30,day) calculated assuming no de novo, i.e., light-dependent emissions? Even though, MT emissions at least are assumed to be largely light-dependent?

This graph appears to show extremely high SQT and OSQT emission potentials for a single day, May 18, after which emission potential drops throughout most of the early growing season. Given this extreme within season variability, is it reasonable to lump all seasonal data together to arrive at a seasonal mean emission potential?

Why are bud break data shown as below detection limit when values are reported in Figs. 2 and 3?

I can't help but be struck by the following. Up to 15 May, all emissions are apparently below detection limit, while three days later emissions of all terpenoids have been initiated, with SQT and OSQT emissions at their seasonal high. It is a shame that measurements weren't conducted over the intervening 3 days to better understand the onset of emissions and the possible roll of storage pools (although as indicated above, I wonder if some of the reported rates may be artificially high if estimates of leaf biomass is underestimated in these small and rapidly expanding leaves).
* * *

---

## Author Comment (AC1) · 23 Mar 2021

Thank you for the valuable comments and corrections! We have considered them carefully and modified our manuscript accordingly. Please, see below the detailed answers to the comments.

This manuscript provides a large and useful dataset describing seasonal variation in BVOC emissions from Betula pubescens, the most important deciduous tree species of the Eurasian boreal forest. Of particular significance is the inclusion of a number of previously understudied compounds, sesquiterpenes and oxygenated sesquiterpenes, which are expected to play an oversized role in atmospheric chemistry due to their potentially high rates of reactivity and SOA formation. Inclusion of such reactive compounds may help to reconcile discrepancies between leaf level estimates of OH reactivity and measurements of OH reactivity in the forest atmosphere. The authors have made a large number of repeated measurements on single branches across each of two growing seasons and supplemented their data set with qualitative BVOC emissions characterizations of branches of an additional 13 trees. As a consequence, they observed a great amount of variability across seasons and between tree individuals, both quantitatively and qualitatively, i.e., the chemical species composition of emissions. This high amount of variability necessarily results in a messy, somewhat confusing dataset and complicates data interpretation, but documenting the variability is itself an important result. And despite this variability, the authors were able to draw fairly robust general conclusions about seasonal trends and the relative importance of different classes of emitted BVOC over time. While I regard these emission rate data and the conclusions drawn as qualitatively valid and worthy of publication, I have a number of reservations regarding the quantitative validity of their measurements and their calculated BVOC emission factors, as outlined below. Apparently, and inevitably when a singe branch is measured repeatedly over an entire season, leaf biomass could not be measured until the end of the experiment. Therefore, biomass was necessarily estimated, in this case using a growth model developed for needles of Scots pine. Its validity for deciduous species is unaddressed. While estimates for mature leaves, when growth has slowed or stopped are likely to be quite accurate, biomass estimates during bud break and early leaf growth seem highly problematic, and errors could lead to corresponding errors in emission estimates. Given the observed high emission rates of SQT and OSQT immediately following bud break, one might wonder if underestimates of leaf biomass contribute to these high rates. Was there any effort to determine early season leaf biomass, specific leaf mass, etc. on comparable nearby branches or to validate the CASSIA model for B. pubescens?

-We did measure biomass of leaves three times over the growing season. However, as mentioned by the reviewer biomass during the bud break and early growth was modelled. Measurements of birch leaf growth were conducted at our site during years

2015 and 2016 throughout the summer beginning immediately after bud break, so the model describes the leaf growth from the beginning. Measurements included both area growth measured from photographs and specific leaf mass (g/mm2). Thus, the modelled mass growth used in this manuscript took into account both the area increment and the changing mass to area ratio. Correlation between the modeled and measured leaf growth was good during 2015 and 2016 and the modelled results seemed reasonable based on visual check when compared with the photographs taken during the study period of this manuscripts. We improved these explanations in the revised manuscript.

Branch enclosures lacking any sort of temperature control or within chamber mixing, relying solely on air flow through the chamber, inevitably experience above ambient temperatures under sunlit or partial sun conditions. The authors address this issue, characterizing the difference between ambient temperatures inside and outside the enclosure, but the over temperatures in the chamber are enormous, averaging 10 to 14 deg. under partly cloudy and sunny conditions, respectively. Given typical Q10 values for BVOC emissions, this would likely result in 2 to 4-fold increases in branch emissions within the chamber compared to those outside the chamber. Thus, reported emission rates should be viewed with some skepticism. Of course, the relevant temperature for characterizing emissions is the leaf temperature, rather than chamber air temperature. Using an infrared thermometer, the authors characterize the relationship between air temperature and leaf temperature on branches outside the chamber. As expected, these data suggest that sunlit leaves outside the measurement enclosure experience temperatures significantly above ambient, as much as 8 deg. in sunny conditions (which seems a little high). The authors use this data to suggest that, since leaves outside the chamber experience temperatures significantly above ambient air temperature, the elevated air temperature within the enclosure is not so problematic. This ignores the possibility, however, that leaf temperatures within the enclosure are also significantly elevated above chamber ambient air temperatures under partial or full sun conditions, further exacerbating the overtemperature problem. While the authors maintain (p. 5, l. 22) that a flow rate of 3.0 lpm through a 6 liter enclosure was sufficient "to keep the leaf temperature close to the chamber air temperature" they offer no evidence in support of that statement. One wishes the authors had placed thermocouples on leaves inside the chamber, i.e., leaves actually measured, ideally throughout the course of the experiment, but at the very least in order to validate their assumptions regarding within chamber leaf temperatures.

-We understand the reservations of the reviewer and his arguments are valuable. However our most important aim in the measurements presented in this manuscript were to capture SQTs, OSQTs and DTs. As these compounds are normally present at low concentrations and are very easily lost on surfaces, we chose to reduce all surfaces to a bare minimum and had them inert (FEP). In addition, the dilution was chosen to be rather small, to maximize the measured signal. The used setup allowed us to measure SQTs and OSQTs, but still was not sensitive enough for DTs. The challenge of measuring these compounds can also be seen in Table 3 (as the reviewer mentions in a later comment). Unfortunately, this setup has its drawbacks, as mentioned by the reviewer, and we are aware of its limitations (i.e. chamber PAR, surface temperature, limited cooling). Especially with the cooling we could (so far) not find a good compromise that would allow to properly cool the chamber without diluting the signals too much or adding additional surfaces to the setup. We understand that these limitations are adding errors to our measurements and reported emission coefficients, however we are also quite certain that with all the suggested additions (PAR inside the chamber, surface temperature, high dilution for cooling) we could not have acquired the presented data. We see our results as a first estimate, and hope that in the future we or other researchers can/will develop better setups to measure these evasive compounds together with the suggested parameters.)

- The 3 lpm flow through the chamber was just the minimum. We changed now our manuscript to reflect the fact that the flow was 3 to 7 lpm and that in 2019 it was always >6 lpm

- We also now state the temperature effect more clearly in the first paragraph of results (section 3.1).

In a related issue, the use of PAR, measured above the enclosure, is of limited utility when trying to characterize the actual irradiation on a multitude of leaves in a branch enclosure, where leaf angle and self-shading can drastically reduce the light reaching an individual leaf. For a certainty, the average PAR received by leaves within the enclosure is significantly less than that measured above the chamber. Since at least some of the BVOC emissions result from de novo production, dependent on light, inadequate characterization of PAR represents a problem, particularly when trying to apply the Guenther et al. light algorithm to estimate emission potentials, discussed further below.

- This clearly is an issue and the uncertainty of PAR measurements is now better stated in the manuscript both in the methods and results sections. The reason for not having a PAR sensor inside the chamber is that we had to avoid all active surfaces in the chamber to be able to capture also the emissions of higher terpenes, which are very easily lost on the surfaces as shown by e.g. Helin et al. (2020). We always stress the importance of conducting additional ecosystem level measurements, which overcome the difficulties in temperature and radiation measurements. However, when concentrating in very reactive compounds like in the current study, the enclosure measurements are the only option. We have added this issue in our conclusions.

With respect to de novo emissions versus emissions from storage pools, the analysis presented here is somewhat confusing. Citing previous studies employing 13C to identify de novo production of monoterpenes in birch species or studies where emissions fell to zero upon chamber darkening, the authors quite reasonably assume that some or all of the monoterpenes measured in this study would evidence a light dependency. If one assumes that most or all of MT emissions arise de novo, why solve for solve for emission potential using only a temperature function (Table 3)?

- As presented in Table 3, using the light dependent algorithm did not improve the correlation. With only the temperature we were able to represent emissions over the growing season with relatively good confidence, so that only the temperature would be required to upscale these emissions in atmospheric models if no PAR data is available. The temperature dependent algorithm was also used to enable a comparison with earlier studies which mainly state only the temperature dependent emission potentials.

A simple darkening of their enclosures would have removed any doubt, as well as providing information regarding light dependency of other classes of BVOC, such as SQT and OSQT, some of which (including B-caryophyllene, B-farnesene and linalool, all significant emissions in this study) are at least partially the result of de novo production in other tree species (e.g., Ponderosa pine (Harley et al. 2014), although Hakola et al. (2001) report little effect of darkening on SQT emissions in B. pubescens.

- As mentioned we have done darkening experiment of B. pubescens in the earlier study (Hakola et al. 2001) and we still rely on those results. However, in future studies we will repeat this as suggested by the reviewer.

Although the authors assume that the observed SQT emissions arise from storage pools, and are therefore dependent on temperature alone, they nevertheless present data in Table 3, calculating SQT (and OSQT and ALD) emission potentials assuming both light and temperature dependencies.

- This was done to show that taking light into account did not improve the correlation.

If one examines Fig. 3, the lack of a light dependency for SQT and OSQT emissions is indeed called into question. If one compares the early season data for MT, SQT and OSQT emissions, the shape of the responses is almost identical. That is, all show very low (but not zero) nighttime emissions, followed by a more than 10-fold increase during the day. This large an increase is very unlikely to be explained simply by a 17 deg temperature change, requiring a Q10 of over 5. With respect to MT, this large increase is easily explained by including a light response. But what about SQT and OSQT?

Again, a simple darkening of the enclosure would have helped resolve the issue.

- For less volatile STQs and OSQTs the temperature dependence is expected to be higher than for MTs due to their lower vapour pressures. As mentioned earlier, we also relied on our earlier darkening results where sesquiterpenes emissions did not decrease significantly during darkening.

Throughout the MS, seasonal means of emissions are reported. If I understand correctly, these are simply the means of all measurements made during a given season. This has some information value. But since the majority of measurements were apparently made during periods of darkness or low light and low nighttime temperatures(using data points in Fig. 3 as a guide), seasonal means significantly underestimatemean midday maximal emissions. For example, the early seasonal mean of all SQTemissions is 692, while the midday maximum in Fig. 3 is well over 2000. Using sea-sonal means of all data seriously underestimates the significance of midday emissions,of greatest importance for most atmospheric chemistry issues. Nowhere in the MS are mean daily maxima of emissions reported. Similarly, reporting only seasonal means ofPAR (including nighttime PAR) and temperature is of modest values; reporting daytimemeans or maxima would be of more use.

- We now added afternoon means to sections 3.1.2 and 3.1.3 and to the appendix table A1. In addition, a comment on this was added into the reformulated abstract. In Fig. 2, standard deviations of the measured emission rates are also shown.

Given the issues related to leaf temperature and PAR incident on leaves, I have very little confidence in any of the emission potential estimates provided in the manuscript. In addition to the temperature and light issues, I have some issues with the general procedures used to determine emission potentials. If I understand correctly (the au-thors don't present much detai) all of the data in a given season are lumped together and the data are fit to either the Guenther temperature algorithm or the light and temperature algorithms. I am unsure how ppropriate, or useful, this is, particularly if the majority of

the data is obtained under conditions of darkness and relatively low temper-ature. If I've misunderstood how these emission potentials are arrived at, I apologize,but more detail about the procedure might be warranted. Finally, given the consider-able variability in emissions even within a given season (Fig. 6), I question the utility of publishing seasonal mean values. The utility of calculating these emission potentials by the methods employed is further called into question by Table 3, in which almost half of the determinations are not considered sufficiently robust to report, while others generate unrealistic estimates of Beta.

- Emission potentials are mainly calculated for atmospheric modelling. For these models there is little benefit for daily emission factors since it is not possible to include very detailed data. In addition, for calculating daily emissions potentials, there is only a few measurement points, which may lead to deviations in the potentials. Therefore seasonal mean emission potentials are more representative for the use in modelling even though, at the moment, even seasonality is not often taken into account in these models. Often emission studies are only from short campaign or only from a few manual samples taken over the growing season and seasonality is not detected that easily.

- Unrealistic values in Table 3 are due to the low amount of values above the detection limits of our instruments. Thus, the values detected being very low and close to the detection limits have high uncertainties. We added a note about this to the caption of the table. Some of the compounds were measured for the first time exactly because the values are low as these compounds are usually lost easily to surfaces.

Thus, I recommend that the authors focus more on the measured emission rates and less on the attempts to determine emission capacities. They should present their emission data, being straightforward about the enclosure temperature issues and the resulting impact on their measurements. Having done so, they can still draw reasonable conclusions regarding the suite of BVOCs emitted by B. pubescens, the seasonal and tree to tree variability, and about the importance of previously unreported OVOC or OSQT emissions.

- We still hope to present the emission potentials (even with these uncertainties) since without parameterizing emissions somehow it is not possible to estimate their possible impacts in the atmosphere using atmospheric models. As shown by the emission algorithms, there is a clear correlation with temperature, and even with uncertainties measuring the temperature, these can give a first estimate on the possible emissions. Hopefully, in the future, we have better methods (maybe ecosystem scale flux measurements) to estimate emissions of these highly reactive compounds with strong potential for aerosol production in the atmosphere as well.

A few other issues require some attention, as follows.

Abstract, line 24. See comments above regarding the reporting of seasonal means).

The abstract was reformulated with a comment on afternoon maxima.

Fig. 1. Early dates in 2017 should be 24-28/5 (although I find 24-28 May preferable)

- The dates on the photos are correct. They simply do not always match with the measurement days. Unfortunately, we do not have photos from all days, since they were taken during site visits, which happened a few times each month. Nevertheless, these photos show the development of the leaves.

Reconcile p. 5, line 12 in which losses are "negligible" with p. 15, line 19 in which significant (?) losses are implied for high MW OSQTs.

- We clarified this in the manuscript by changing the sentences on p. 5 to 'For most target compounds losses in these sampling lines and chamber are expected to be negligible as demonstrated by the acceptable recoveries observed in the laboratory tests (Helin et al., 2020; Hellén et al., 2012), and since high flow rates were used. Even though the only OSQT (caryophyllene oxide) studied had also acceptable recovery (>80%), some losses of higher molecular weight compounds (i.e. diterpenes) in the chamber were detected and therefore it is possible that there are some losses of higher OSQTs also in the current study and our OSQT emission rates are underestimated.'

and on p. 15 to 'In our earlier tests we have detected some losses of higher molecular weight compounds into our chamber and particularly in the instrument (Helin et al. 2020).'

p. 2, line 35. "These OSQTs are expected to be highly reactive and to have higher-comparative secondary organic aerosols yields. . ." Likely true, but please provide areference.

- To our knowledge, there are no studies on the SOA yields of OSQTs, but we added into the manuscript that this assumption was based on the larger size and lower vapour pressure of these molecules.

p. 4, line 15. How long was the branch enclosed in the chamber during measurements?How soon after enclosing the branch was sample collection begun?

- In 2017 the measured branch was enclosed for 1 to 2 weeks. In 2019 same branch was enclosed before the bud brake and during the early growing season measurements between 6/5-7/6, and after that for 1-2 weeks at the time. After closing the chamber sample collection started immediately, but results from first samples were removed. This clarification was added to the manuscript.

p. 5, line 9. What was the sampling duration? I.e., how large a sample collected?

- We have described this already on page 6 line 16-17 and therefore we do not repeat it here.

p. 5, line 22. I'm not convinced that a flow rate of 3 lpm through a 6 liter enclosure issufficient to ensure that leaf temperatures remain close to enclosure temperature. Dothe authors have evidence? Measuring temperature of leaves outside the enclosuredoes not adequately characterize leaf temperatures within the chamber. Why wereleaf temperatures not measured on leaves inside the enclosure, ideally on a continuousbasis, but certainly for brief periods to test these assumptions under varying levels ofirradiance?

- Unfortunately, we did not have any system to measure leaf temperatures inside the chamber since our surface thermometer was not able to measure through the FEP-film of the chamber. In future studies we will set up a system to measure leaf temperature. Of course this will also give an estimate of one or a few leaves while the leaves in the chambers are in very variable light conditions and we need to be very careful with the materials since SQTs and OSQTs are very easily lost on the surface materials. So far, we have not found a good compromise for that problem. However we still hope that the presented results have their value, even with the additional errors.

p. 6, line 11. Empty chamber blanks were subtracted. Please give some idea of themagnitude of these corrections. Can they be considered negligible?

- For terpenes, blank was negligible, but for aldehydes in 2017 it was between 3 to 7 ng g-1 h-1. This was added to the manuscript.

p. 7, line 16. Since you are "solving" for E30, shouldn't the form of the equation be:

E30 = E/(exp(Beta*(T-Ts)))

Of more significance, what value of Beta was used, or were Beta and E30 solvedfor simultaneously? It appears from Table 3 that they were solved for simultaneously,with values of Beta ranging from 0.03 to 0.15. Guenther 93 reported a range of Betavalues for different species varying from 0.057 to 0.144 and uses a mean value ofBeta=0.09. I see no reason why the temperature dependency of emissions for a sin-gle species should vary so widely, and in the absence of experimental determinationof Beta (emission measurements while varying leaf temperature under otherwise con-stant conditions), I think choosing a constant value for Beta would be the wiser course.

- The equation was reformulated as suggested by the reviewer. Beta was solved simultaneously. In earlier studies it has been shown that temperature sensitivity of emissions maybe be higher in northern areas and there can be differences even with the same tree during the season and due to stress. In addition, Beta=0.09 would be valid only for

MTs and their emissions were minor. For SQTs and OSQTs with lower vapour pressures, Beta-values are expected to be higher, however, there are no standardized beta values for them (most probable due to low amount of available data). For clarification we added to the manuscript that Beta was solved simultaneously.

p. 8, line 20. "This was the case particularly in SQT and OSQT emissions." It's not clear to me what this sentence refers to. Are you implying that SQT and OSQT are primarily released from storage pools? Based on what evidence? The daily pattern of emissions of MT, SQT and OSQT in Fig. 3 are almost indistinguishable (discussed further above).

- We removed this unclear sentence from the manuscript

Fig. 3. Please indicate whether these are plots of individual days within each season or plots of seasonal means? If individual days, which of the days shown in Fig. 6 (in which emission potentials varied widely within a season) was chosen? Is it representative, i.e. typical daily behavior?

-Seasonal means were used and this was now clarified in the figure caption.

p. 10, lines 3 and 22. As I've indicated, I question the value of reporting mean emission rates by season. However, if you choose to do so, saying that "seasonal mean emission rates were 5 – 690" conveys very little information and is actually confusing. Something like "Mean emission rates in 2017 were significantly higher in May (692) than in June-July (226) or August (5). Similarly, in 2019. . ."

- We changed these sentences as suggested by the reviewer to 'Mean emission rates in 2017, when measured tree was growing in the pot, were significantly higher in early (692 ng gdw-1 h-1) than in main (226 ng gdw-1 h-1) or late (5 ng gdw-1 h-1) growing season. Similarly, in 2019, when a naturally growing tree was measured, mean SQT emission rates in early season were 505 ng gdw-1 h-1, while in in main or late season they were only 41 and 14 ng gdw-1 h-1, respectively.'

p. 11, line 8. As discussed, GLV emissions are clearly associated with leaf damage/ stress. Given that, seasonal mean values are more or less meaningless. I suggest eliminating Table A2 and stressing the highest values of GLV emissions as representing stress or damage responses.

- Good remark. We removed these for GLVs.

p. 13, line 19 and elsewhere. Was OSQT9 identified as 14-hydroxy-_-caryophyllene acetate or not? If so, refer to it by name; if not, say "tentatively identified as . . ."

- We did not have an authentic standard for it and therefore it was only tentatively identified and the manuscript was corrected accordingly.

p. 15, line 5 "Emission potentials showed a very high variation between the seasons." This is true, but emission potentials also showed large variation within seasons (Fig.6).

- We changed the sentence accordingly

Fig. 6. Are these emission potentials (ET30 or E30,day) calculated assuming no de novo, i.e., light-dependent emissions? Even though, MT emissions at least are as- sumed to be largely light-dependent? This graph appears to show extremely high SQT and OSQT emission potentials for a single day, May 18, after which emission potential drops throughout most of the early growing season. Given this extreme within season variability, is it reasonable to lump all seasonal data together to arrive at a seasonal mean emission potential?

- These are only temperature dependent emissions. We tested also light and tempera- ture algorithm but it did not give any better R2 values. Also with light and temperature algorithm, the emission potentials were high on 18 May. These emission potentials could be due to budbreak. It has been shown earlier that budbreak causes high emis- sions possibly due to stored compounds in the buds. We think that even though there are strong daily variations, it was still clear that emissions were clearly higher during the early growing season, which is also shown by these seasonal means. When modelers are using emission potentials in atmospheric models, they most often use only one emission potential for the whole year and here our aim was to show that there are strong variations between the season and emissions may vary, especially in early growing season. To make this very clear, we are hoping to show also these seasonal averages. Very often emission measurements are based only on a few samples or on short (e.g. 2 -4 week period) during the main growing season, but here we had in situ GC-MS measurements over the whole growth period.

Why are bud break data shown as below detection limit when values are reported in Figs. 2 and 3?

- There were not enough data points for the calculation of temperature dependence. This is now clarified in the caption of Figure 6.

I can't help but be struck by the following. Up to 15 May, all emissions are apparently below detection limit, while three days later emissions of all terpenoids have been initiated, with SQT and OSQT emissions at their seasonal high. It is a shame that measurements weren't conducted over the intervening 3 days to better understand the onset of emissions and the possible roll of storage pools (although as indicated above, I wonder if some of the reported rates may be artificially high if estimates of leaf biomass is underestimated in these small and rapidly expanding leaves).

- We are also sad to have missed this crucial moment. We intended to measure continuously over the summer, however, these measurements are very challenging and there are often problems and malfunction with the instruments. Unfortunately one malfunction happened exactly during that time.

---

## Author Comment (AC2) · 23 Mar 2021

Thank you for the valuable comments and corrections! We have considered them carefully and modified our manuscript accordingly. Please, see below the detailed answers to the comments.

General comments: The chemical identities, surface fluxes, and biological and environmental dependencies of volatile organic compound emissions into the atmosphere from downy birch trees is an important research topic for understanding the biology of these widespread tree species in higher latitudes and their interactions and feedbacks with the atmosphere. It is particularly nice to see a VOC emission study based on GCMS, as compound specific information is obtained due to the separation of isomers

(e.g. monoterpenes). I particularly found the observations of oxygenated sesquiter-penes to be extremely exciting and new, as they are rarely reported as emissions from plants. I would suggest the title of the article and the paper itself focus on this particu-larly novel finding. However, the authors need to discuss the limitations of the GC-MS technique as many compounds were likely missed, especially the low molecular weight OVOCs like alcohols, aldehydes, ketones, acids, esters, etc. Some of the strong con-clusions mentioned in the paper would likely not stand if authors also used a method that is good for these compounds (e.g. PTR-MS).

- We added a discussion on the limitations and advantages of the GC method.

The paper is also written as if birch trees in the natural forest were studied, but reading the methods, it is clear that only a potted birch tree was used in one year and then a natural birch tree a second year.

- We now clarified this in the manuscript

In general, it is not clear to me why the authors did not select several natural birch trees growing near the instrument container to study. It's hard to extrapolate data from a single tree as being representative of downy birch emissions from the Boreal forest.

- It was not possible since we had to keep inlet lines as short as possible so that we do not lose SQTs and OSQTs in them and there was only one birch growing close enough to our measurement container.

In general, the comparison of the two datasets from different years, one using a natural tree and the other using a potted tree, and with different experimental methods rather awkward. This if further of concern since the flow rates to the instruments were different between the years. I would perhaps suggest picking only one of the data set years and focus the article on just that (perhaps moving the other to supplementary material).

- Often these kind of emission measurements are done only for a short period (e.g. two week in mid-summer) or using offline sampling with only a few samples. Here we

wanted to show how much emissions may vary in different setups/seasons. Measurements from the year 2017 were included to show the effect of stress on emissions. We now clarified this in the text. The flow rates to the instruments should not affect the results since it is taken into account and it is assumed that there are no significant losses of these compounds in the inlets when using these flows. We have studied inlet recoveries in Helin et al. (2020) and Hellén et al. (2012).

The measurements were collected during different developmental stages, but the impact of developmental stage on the emissions is poorly treated in the paper. Also with respect to discussions on stress, visible signs of damage, drought, temperature, etc., it is not clear why these occurred and when and how they impacted the emissions.

- Stress occurred only in 2017 when the measured birch was growing in the pot, due to dry zero air and too little watering. In 2019, the zero air used was humidified and the tree was growing naturally. The effect of stress in 2017 was confirmed by the strong emissions of GLVs which are known to be stress related compounds. This clarification was added to the manuscript.

Of particular concern is the fact that the study did not measure key physiological processes/variables to indicate that the vegetation was physiologically active including transpiration, stomatal conductance, and net photosynthesis rates.

- Unfortunately, these measurements were not available (see also answer to Q2 of reviewer 1). However, since most often these emissions are modelled in the atmospheric models only using temperature, they are not needed for using these results.

In particular, the use of dry zero air likely caused stomatal closure due to VPD stress, likely associated with high temperatures as well.

- We did measure RH in the chamber. In 2017 the mean RH was 31% while in 2019 it was much higher (63%) due to humidification of the zero air used for flushing the chamber. We added a comment on the possible stomatal closure in 2017 into the

manuscript in section 3.4

Regarding temperature, why was it not measured always and why was a branch outside the enclosure measured instead of the actual branch being studied inside the enclosure?

- Unfortunately we did not have any system to measure leaf temperatures inside the chamber since our surface thermometer was not able to measure through the FEP-film of the chamber. We did measure chamber temperature continuously. In addition, we have to be very careful with the materials since SQTs and OSQTs are very easily lost on the surfaces and therefore we tried to avoid installing any additional equipment inside the chamber. We assumed that zero air flow into the chamber was enough to keep leaf temperature close to the chamber temperature. Also with some thermocouple for leaf temperature measurements, one would measure only one or a few leafs while other leaves in the chamber in different light conditions may have very different temperatures. Getting the correct leaf temperature measurements in these kind of measurements is challenging, but in future we hope to get a better system for this.

The paper does report diurnal patterns of emissions which in my opinion is more important the growing season.

- We report both patterns, as their significance depends on interest of the reader. In case the interest lies in physiological processes, the diurnal pattern will be more informative, however, if the interest is in global modeling, the seasonal pattern will be of more use. Our intentions were aiming more to global modeling (as can be seen by the absence of key parameters for physiological processes), still we hope the results can be helpful in both areas.

Specific comments

Title: The title is perhaps misleading as the study did not measure all possible VOCs from the trees, and it does not indicate which tissue the emissions are deriving from

and the environmental/biological conditions.

- It is never possible to measure all VOCs simultaneously, as estimations go to over 1000000 VOCs with less than 11 carbon atoms (Goldstein et al. 2007). Terpenes and terpenoids clearly dominate VOC emissions (Sindelarova et al. 2014). We covered major SOA precursors while missing lighter more volatile compounds. To clarify we changed the title to 'Sesquiterpenes and oxygenated sesquiterpenes dominate the VOC (C5-C20) emissions of downy birches'.

Goldstein, A. H. and Galbally, I. E.: Known and unexplored organic constituents in the Earth's atmosphere, Environ. Sci. Technol., 41, 1514–1521, https://doi.org/10.1021/es072476p, 2007)

Sindelarova et al., Global data set of biogenic VOC emissions calculated by the MEGAN model over the last 30 years, ACP, doi:10.5194/acp-14-9317-2014

Could the dominant VOCs change under drought or high temperature stress?

- Yes, huge amounts of GLVs are emitted due to stress also in this study. In addition, for example, farnesene is known to be related to stress emissions. This is discussed in the manuscript in section 3.4.

What about from stems or roots as apposed to leaves?

- Stem and root emissions are expected to be low compared to needle/leaf emissions (Vanhatalo et al. 2020 and Mäki et al 2019).

Vanhatalo A, Aalto J, Chan T, Hölttä T, Kolari P, Rissanen K, Kabiri K, Hellén H and Bäck J (2020) Scots Pine Stems as Dynamic Sources of Monoterpene and Methanol Emissions. Front. For. Glob. Change 2:95. doi: 10.3389/ffgc.2019.00095.

Mari Mäki, Hermanni Aaltonen, Jussi Heinonsalo, Heidi Hellén, Jukka Pumpanen & Jaana Bäck: Boreal forest soil is a significant and diverse source of volatile organic compounds. Plant Soil, https://doi.org/10.1007/s11104-019-04092-z, 2019.

https://link.springer.com/article/10.1007%2Fs11104-019-04092-z.

What about the phenological dependence of VOC emissions from leaves as has previously been described for broadleaf species? https://acp.copernicus.org/articles/16/6441/2016/

- Our manuscript describes the seasonality of different terpene/terpenoid groups. We have divided the emissions to seasons: budbreak, early and late growing season. One of our conclusions is: "Due to high variability of the emissions over the growing season, it is clear that estimating birch emissions should take into account the seasonality of emission potentials"

Abstract: The abstract is very long and lacks critical structure (suggest the authors review a widely accepted format for abstracts and adhere to it) http://www.cbs.umn.edu/sites/default/files/public/downloads/Annotated_Nature_abstract.pdf I am not really following the motivation of the study. It seems that little data is not really something that is scientifically motivating. Why not start with a broader perspective discussing key uncertainties of VOC emissions from the Boreal forest and the missing OH reactivity problem?

- We reformulated the abstract.

What does, "almost throughout the summer" mean?

- We removed this and state now that SQTs and OSQTs were the main emitted 'terpenes' (and not 'compounds')

Are mean emissions the most useful here? I would suggest reporting emissions separately for bud break, early, late, and main growing season.

- This was corrected

Is bud break supposed to be different from early growing season?

- Yes, there was a clear difference. During the budbreak period, when there was a

bud, emissions were very small, and when the leaf started to grow fast (early growing season) emissions increased a lot. A better description of the various periods has been added to the section 2.1.

If no significant isoprene emissions, I would probably not mention that in the abstract.

- We removed isoprene from the abstract.

"Variable levels of emissions of MTs, C5-C10 aldehydes and GLVs were detected". Variable with what?

- This sentence was removed.

Why are emission values of isoprene listed (below detection limits), but emisisons of these other compounds are not explicitly listed?

- Isoprene data was removed.

"On average SQT and OSQT emissions were 5 and 6 times higher than MT emissions, but variation over the growing season was high and during the late growing season MTs were the main compound group emitted." Do the authors mean total monoterpene emissions and total SQT+OSQT emissions? It is not clear here.

- This was clarified.

"Of the SQTs, _-caryophyllene and _-farnesene were the main compounds emitted in 2019, while in 2017 also high, possibly stress-induced emissions, of _-farnesene were detected." It is clear that there are interannual variability in the emissions, but this sentence does not summarize the key differences.

- Abstract was reformulated.

Regarding stress induced emissions, what is the stress? How do we know that this represents natural emissions and not from an artificial stress by the measurement methods? "due to drought and high chamber temperature". Was temperature in the

chamber not controlled? High enclosure temperatures are often artifacts of many enclosure studies due to improper temperature control. Do the authors consider these emissions natural or artificial high temperature stress emissions?

- The stress was 'artificial' since in 2017 the birch was growing in a pot and, in addition, dry zero air was used for flushing the chamber. In 2019 a natural tree and humidified zero air were used and we did not detect stress related compounds. This has now been clarified in the manuscript.

If the OSQTs are being tentatively identified, how certain are the authors about the reported identity of any of the compounds?

- As explained in section 2.3, they were tentatively identified based on the comparison of the mass spectra and retention indexes (RIs) with NIST mass spectral library (NIST/EPA/NIH Mass Spectral Library, version 2.0). This is a standard method for identifying compounds in GC-MS analysis. This identification should then be confirmed by authentic standards. However, in this case these standards were not commercially available. Based on our own experience, this tentatively identification is usually correct, but sometimes it could be some very similar compound, for example a cis- or trans- isomer. We would say that the identification is much more certain than in for example in PTR-MS analysis. Here we really separate different compounds (also isomeres) in the chromatograph and all individual compounds have specific EI mass spectra and retention index.

What happened during the last two days of the experiment that is different from the other days?

- This is discussed in section 3.1.1. It could be related to the senescence of the leaves.

How does the composition of monoterpenes change throughout a day and season as a function of temperature? How does this correspond to previous studies on broad leaf monoterpene emissions as a function of leaf temperature?

https://onlinelibrary.wiley.com/doi/full/10.1111/pce.12879

-Our monoterpene emissions were very low and often below detection limits. Therefore, we studying their changes in more detail is not justified in our opinion.

Introduction "Monoterpenes (MTs) and sesquiterpenes (SQTs) are the major biogenic volatile organic compounds (BVOCs) emitted from the boreal forest." Why have the authors ignored oxygenated VOCs which have very high emission rates from not only the Boreal forest, but many other ecosystems as well. I understand that the current study if focused on MTs, SQTs, and GLVs, but the lack of inclusion of oxygenated VOCs like low molecular weight acids, esters, alcohols, aldhehyes, ketones, etc. is not justified.

- We corrected this by adding at the beginning 'in addition to oxygenated VOCs and isoprene'. In the third paragraph of the introduction there is also more information on OVOC emissions. In addition, we state now better in the experimental section that our measurements were missing these VOCs.

Very long and confusing sentence, please revise: "A recent study (Praplan et al. 2019) showed that currently known oxidation products are able to explain only a minor fraction (< 4.5%) of the missing reactivity in the air of boreal forest and large fractions of missing reactivity, which was not explained by isoprene, MTs or SQTs, have been found directly in the emissions of main boreal tree species (Nölcher et al. 2013, Praplan et al., 2020)."

- This was reformulated.

The intro seems to conclude that the "missing OH reactivity" are not due to sesquiterpenes or monoterpenes, yet the authors do not convincingly demonstrate this with references. These measurements are highly technical and often a particular GC-MS configuration is not suitable for all of them. Thus, I feel that this strong statement is not justified.

- This conclusion was made by Praplan et al. (2019). The GC-MS set up used by

Praplan et al. (2019) was able to detect all monoterpenes and sesquiterpenes. The GC-MSes and the OH reactivity measurements used the same inlet line and it was confirmed that these terpenes are not able to explain the missing reactivity. Our GC-MS cannot measure oxygenated compounds having more than one or two oxygen atoms and for example nitrogen containing compounds, but is able to detect all hydrocarbons in the range having 5 to 15 (or 20) carbon atoms with detection limits $\sim$1-10 ng/m3.

I also am not confortable with the description in the intro that low molecular weight OVOCs have been well characterized from the Boreal forest. A handful of published measurements is far from being well characterized.

- This was changed to 'have been characterized'.

I would include more background information on the biology and distrubtion/abundance of downy birch (Betula pubescens) across the world. It would be great if the authors could take more of a global perspective of their results, putting it into perspective with other ecosystems, rather than the focus on only Finland.

- More information with wider perspective on downy birch was added.

It's better to remove any phrases about, "this is the first time. . .."

- This was removed.

From the paper. If written well, these statements will not be necessary, and can instead be reserved for the letter to the editor.

Methods Please describe the Carbon number range that the experimental setup is able to quantify by GC-MS. It is likely that all VOCs lower than 5 carbon atoms were missed or poorly quantiative due to breakthrough by this technique, especially with the warm temperature of the cold trap (25 C).

- Yes this is true and we discarded all their results since they were not quantitative. However, with our method it is possible to detect all compounds having a vapour pressure corresponding to alkanes with 5 to 20 carbon atoms. From an atmospheric point of view, these compounds are highly interesting compared to lighter compounds due to their high reactivity and strong potentials to form SOA compared to, for example methanol, formaldehyde or acetone, which are also known to be emitted by many plants. With a PTR-MS you would be able to detect light OVOCs, but you would miss molecular level information on terpenes, which is highly important for estimating their atmospheric impact. Traditional PTR-MS are not very sensitive and quantitative for SQTs (or OSQTs) and it is highly probably that we would have missed them totally. Their measurements would be possible with a Vocus-PTR-TOF, for example, but molecular level information would still be missing. Then, in oder to detect for example ethene, which is known to be emitted, one would need another type of GC. Hakola et al. (1998) have also measured ethene emissions of birches, but they were found to be low compared to terpenes. We added to the manscript the carbon number range measured and comment on missing low molecular weight compounds to the section 2.2

Please show an example chromatogram (GC-MS) of both the standard, blank, as well as an enclosure air sample.

- Examples of chromatograms were added to the appendix.

How was the data on leaf growth rates used in the analysis?

- The model for leaf growth was only giving the estimate of the daily leaf weight and the leaf growth rate was calculated from it. It was used in Fig. 6 to relate the high emission rates with the fast leaf growth.

It is not clear why a separate branch enclosure was used with the results deemed semiquantitative (from a separate tree). It's not clear why this is presented in the paper.

- It was done to show that it is not just our birch which is emitting these newly identified compounds (OSQTs). For example, Scots pines are known to have different chemotypes, which emit different set of monoterpenes. We wanted to confirm if this is the case for the Downy birch. We added some explanation with a reference to paper about the chemotypes of Scots pine (Bäck et al., 2012) to the manuscript.

Bäck, J., Aalto, J., Henriksson, M., Hakola, H., He, Q., and Boy, M.: Chemodiversity of a Scots pine stand and implications for terpene air concentrations, Biogeosciences, 9, 689-702, doi:10.5194/bg-9-689-2012, 2012.